# Hevein-Like Antimicrobial Peptides Wamps: Structure–Function Relationship in Antifungal Activity and Sensitization of Plant Pathogenic Fungi to Tebuconazole by WAMP-2-Derived Peptides

**DOI:** 10.3390/ijms21217912

**Published:** 2020-10-24

**Authors:** Tatyana Odintsova, Larisa Shcherbakova, Marina Slezina, Tatyana Pasechnik, Bakhyt Kartabaeva, Ekaterina Istomina, Vitaly Dzhavakhiya

**Affiliations:** 1Laboratory of Molecular-Genetic Bases of Plant Immunity, Vavilov Institute of General Genetics RAS, 119333 Moscow, Russia; omey@list.ru (M.S.); mer06@yandex.ru (E.I.); 2All-Russian Research Institute of Phytopathology, Bolshie Vyazemy, 143050 Moscow, Russia; beefarmer@yandex.ru (T.P.); kartabaeva040893@mail.ru (B.K.); 3Department of Molecular Biology, All-Russian Research Institute of Phytopathology, Bolshie Vyazemy, 143050 Moscow, Russia; dzhavakhiya@yahoo.com

**Keywords:** hevein-like antimicrobial peptides, antifungal activity, antifungal determinants, synergy, chemosensitization, tebuconazole, plant pathogenic fungi

## Abstract

Hevein-like antimicrobial peptides (AMPs) comprise a family of plant AMPs with antifungal activity, which harbor a chitin-binding site involved in interactions with chitin of fungal cell walls. However, the mode of action of hevein-like AMPs remains poorly understood. This work reports the structure–function relationship in WAMPs—hevein-like AMPs found in wheat (*Triticum kiharae* Dorof. et Migush.) and later in other *Poaceae* species. The effect of WAMP homologues differing at position 34 and the antifungal activity of peptide fragments derived from the central, N- and C-terminal regions of one of the WAMPs, namely WAMP-2, on spore germination of different plant pathogenic fungi were studied. Additionally, the ability of WAMP-2-derived peptides to potentiate the fungicidal effect of tebuconazole, one of the triazole fungicides, towards five cereal-damaging fungi was explored in vitro by co-application of WAMP-2 fragments with Folicur^®^ EC 250 (25% tebuconazole). The antifungal activity of WAMP homologues and WAMP-2-derived peptides varied depending on the fungus, suggesting multiple modes of action for WAMPs against diverse pathogens. Folicur^®^ combined with the WAMP-2 fragments inhibited the spore germination at a much greater level than the fungicide alone, and the type of interactions was either synergistic or additive, depending on the target fungus and concentration combinations of the compounds. The combinations, which resulted in synergism and drastically enhanced the sensitivity to tebuconazole, were revealed for all five fungi by a checkerboard assay. The ability to synergistically interact with a fungicide and exacerbate the sensitivity of plant pathogenic fungi to a commercial antifungal agent is a novel and previously uninvestigated property of hevein-like AMPs.

## 1. Introduction

Despite the successful solution of many plant protection issues by using breeding for resistance and various green biotechnologies [1,2], fungal diseases continue to cause yield losses of economically important crops, including cereals [3], and treatment with chemical fungicides still remains the most relevant way to maintain a favorable phytosanitary situation in the field and to guarantee high yields. At the same time, it is well known that the use of fungicides is associated with ecological risks, and their toxic residues in forage and food products can cause problems for animal husbandry and human health. In this regard, the introduction of ecologically compatible plant protection strategies, including proper control of fungal diseases by minimal effective doses of fungicides, is now the most in-demand tendency in sustainable agriculture. The study of biomolecules with antifungal activity, which might be used as potential biopesticides or sensitizers enhancing the sensitivity of plant pathogenic fungi to modern industrial fungicides in order to reduce their effective dosages, could contribute to the development of integrated disease management technologies meeting the requirements of sustainable agriculture. Plant antimicrobial peptides (AMPs), defense molecules produced by plants for protection against pathogens, are promising candidates for such a study for many reasons.

AMPs represent a structurally diverse group of cysteine-rich molecules suppressing growth and development of pathogenic microorganisms via different modes of action [4,5,6,7,8,9,10]. Of them, plasma membrane permeabilization is the most common; however, some AMPs target intracellular components. Hevein-like peptides constitute an AMP family whose members share structural similarity with hevein, an antimicrobial peptide from the latex of *Hevea brasiliensis* (Willd. ex A. Juss.) Müll. Arg. [11]. Hevein-like AMPs possess a chitin-binding site involved in binding chitin and related oligosaccharides. The mode of action of hevein-like AMPs is poorly understood. It is generally acknowledged that binding chitin and related oligomers of the fungal cell walls mediates antifungal activity [12,13,14]. However, the underlying molecular mechanisms have not been elucidated so far. Earlier, from the kernels of the wheat *Triticum kiharae* Dorof. et Migush., we isolated two unique hevein-like AMPs named WAMP-1a and 1b, which differ by a single C-terminal arginine residue [15]. They contain 44 and 45 amino acid residues, respectively, and are positively charged [15]. We determined the solution structure of WAMP-1a and showed that its compact molecule contains an antiparallel four-stranded beta-sheet, a 3_10_ helix and an alpha-helix [16]. Similar peptides were later discovered in other *Poaceae* species [17,18,19]. Analyzing the biological diversity of *wamp* genes in Poaceae plants, we discovered high conservation of the amino acid sequences of the mature WAMPs with a single variable position 34, in which several amino acids were found: Ala, Lys, Asn, Glu and Val [17,19]. This residue is located in loop 6 (loops are regions between the adjacent cysteine residues) between the alpha-helix (residues 29–32) and beta-strand 4 (residues 36–39) (Figure 1) [16]. WAMPs are synthesized as precursor proteins containing a signal peptide, a mature peptide and a C-terminal prodomain [17]. The features of the WAMP family peptides that distinguish them from other hevein-like AMPs are as follows: (1) a novel 10-Cys motif distinct from previously reported in hevein-like AMPs, (2) a unique structure of the chitin-binding site, in which the conserved serine residue is replaced by glycine which reduces binding to oligosaccharides, (3) similarity of WAMP peptides to the chitin-binding domains of class 1 chitinases at the amino acid and 3D structure levels [15,16], (4) high in vitro inhibitory activity towards pathogens containing and not containing chitin in the cell walls [15,19,20]. Up-regulation of WAMP genes upon pathogen attack was shown pointing to their defense role in planta [21]. We also discovered that WAMPs are specific inhibitors of fungalysin, a secreted metalloproteinase of *Fusarium* fungi that specifically cleaves plant defense chitinases between the chitin-binding and catalytic domains and suggested a WAMP-mediated mechanism, by which plants protect themselves against the deleterious effect of the fungal protease [20]. We also showed that the efficiency of fungalysin inhibition depends on the amino acid residue at position 34 in the polypeptide chain of WAMPs [20]. However, our discovery that WAMPs, which do not inhibit fungalysin, still effectively inhibit different fungi in vitro suggests the existence of an alternative mode of action.

In this paper, to gain an insight into this antifungal mechanism, we report the effect of five WAMPs differing at position 34 on spore germination of different plant pathogenic fungi. To locate the determinants of antifungal activity, we studied the antifungal activity of synthetic peptides derived from the central, N- and C-terminal regions of one of the WAMPs, namely WAMP-2, against these pathogens. In addition, we report here the ability of WAMP-2-derived peptides to potentiate the fungicidal effect of tebuconazole, one of the triazole fungicides of the demethylation inhibitor (DMI) group, which interrupt fungal ergosterol biosynthesis [22], towards five cereal-damaging fungi.

Along with other DMI-fungicides, tebuconazole is highly effective against many fungi that cause harmful diseases of grain cereals grown on an industrial scale [23,24,25]. At the same time, it belongs to the rather persistent xenobiotics [26], the efficacy of which was shown to decrease over time [27,28,29] because of an increasing incidence of insensitive strains in fungal populations. One of the new ways to augment the fungicide efficiency towards such populations, as well as to reduce environmental impact of chemical fungicides in crop-growing areas, is chemosensitization [30,31]. This approach implies applications of fungicides with pathogen-sensitizing compounds (preferably of natural origin) at concentrations at which both fungicides and sensitizers, used separately, induce no or insignificant antifungal effect, but at which pathogen development is strongly inhibited due to a synergistic interaction with each other when conjointly applied [30,31]. The synergy results from the fact that a sensitizing agent targets metabolic pathways in fungi, which are distinct from those targeted by a fungicide [30,32,33,34]. For various natural sensitizers and their synthetic analogues, destabilization of the cell wall/membrane integrity, which is associated with antifungal WAMPs’ activity, is one of the main mechanisms augmenting the sensitivity of fungi to triazoles and other antimycotics. At the same time, WAMPs and tebuconazole attack different targets within the plant pathogenic fungi. In contrast to the cell wall/membrane-disrupting WAMPs, tebuconazole, like other DMI-fungicides, inhibits 14α-demethylase, which is required for the cytochrome P450-dependent oxidative demethylation of 24-methylen-24,25-dihydrolanosterol, a precursor of ergosterol, the major sterol in filamentous fungi [22,28].

Collectively, the current research was aimed at understanding the mechanism of action of WAMP peptides and their role in plant immunity and exploring the potential of using WAMPs and WAMP-2-derived peptides in agriculture to improve the control of plant pathogenic fungi by fungicides in parallel with a reduction of undesirable consequences of fungicidal treatments.

## 2. Results

### 2.1. Recombinant Production and Physicochemical Properties of WAMP Homologues

To study the role of natural variation in the antifungal activity of WAMPs, recombinant WAMPs with substitutions at position 34 were obtained by expression in *Escherichia coli* cells and subsequent purification of the target peptides by RP-HPLC as described earlier [20,35]. The molecular masses of the recombinant peptides were verified by matrix-assisted laser desorption/ionization time-of-flight mass spectrometry (MALDI-TOF MS). Five WAMP homologues were produced: WAMP-1b (Ala34), WAMP-2 (Lys34), WAMP-3.1 (Glu34), WAMP-4 (Asn34), and WAMP-5 (Val34) (Figure 1). We modeled the tertiary structures of WAMP homologues using as a template WAMP-1a, whose solution structure was determined by NMR spectroscopy [16] (Figure 2). The overall compact fold of the molecule is well preserved in WAMP homologues. It consists of four antiparallel beta-strands (2–3, 18–20, 24–26, 36–39) and two short helices (6–8, 29–32) stabilized by 5 disulphide bridges. The WAMP-1a molecule is amphiphilic. The hydrophobic cluster is formed by the side chains of aromatic residues of the chitin-binding site (Tyr-22, Phe-24 and Tyr-31 and two aliphatic residues (Ala-1 and Ala-30) [16]. Residue 34 is located close to the hydrophobic cluster (Figure 2).

### 2.2. Antifungal Activity of WAMP Homologues

The antifungal activity of the recombinant WAMPs was evaluated on the basis of their ability to suppress the germination of spores (conidia) in five plant pathogenic fungi (*Fusarium oxysporum*, *F. culmorum*, *Bipolaris sorokininana*, *Alternaria alternata*, and *Cladosporium cucumerinum*).

All WAMPs inhibited germination of these fungi in the micromolar range, but an evident variation in the antifungal potency among the tested peptides was observed depending on the fungal species (Table 1). Thus, all WAMPs displayed high antifungal activity against *B. sorokininana* (IC_50_ = 22.2–30.6 µg/mL), WAMP-3.1 and WAMP-1b were the most active, while WAMP-2 displayed the weakest activity. The inhibitory activity of WAMPs against *F. oxysporum* was lower compared to *B. sorokiniana*. Again, WAMP-3.1 was the most active followed by WAMP-2.

**Figure 1 ijms-21-07912-f001:**
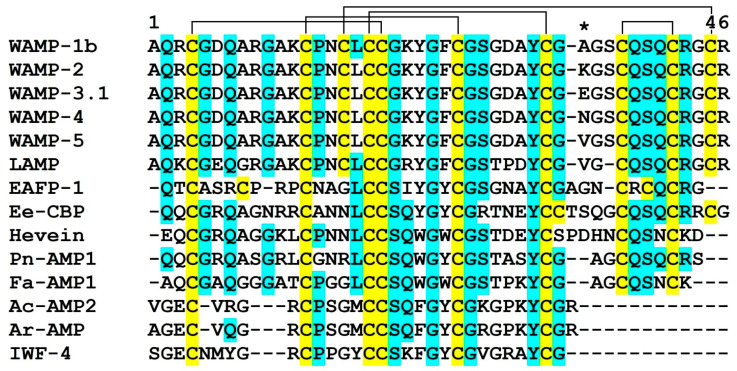
Amino acid sequences of WAMPs and selected hevein-like peptides: WAMP-1b (H6S4F1), WAMP-2 [17], WAMP-3.1 [17], WAMP-4 [17], WAMP-5 [19], LAMP (P86521), EAFP1 (P83596), Ee-CBP (Q7Y238), hevein (P02877), Pn-AMP1 (P81591), Fa-AMP1 (P0DKH7), Ac-AMP2 (Q9S8Z7), Ar-AMP (Q5I2B2), IWF4 [36]. Cysteine residues are shaded yellow, and identical amino acids are shaded cyan. Residue 34 is marked with an asterisk. The arrangement of disulfide bonds of WAMPs is shown above the alignment.

**Figure 2 ijms-21-07912-f002:**
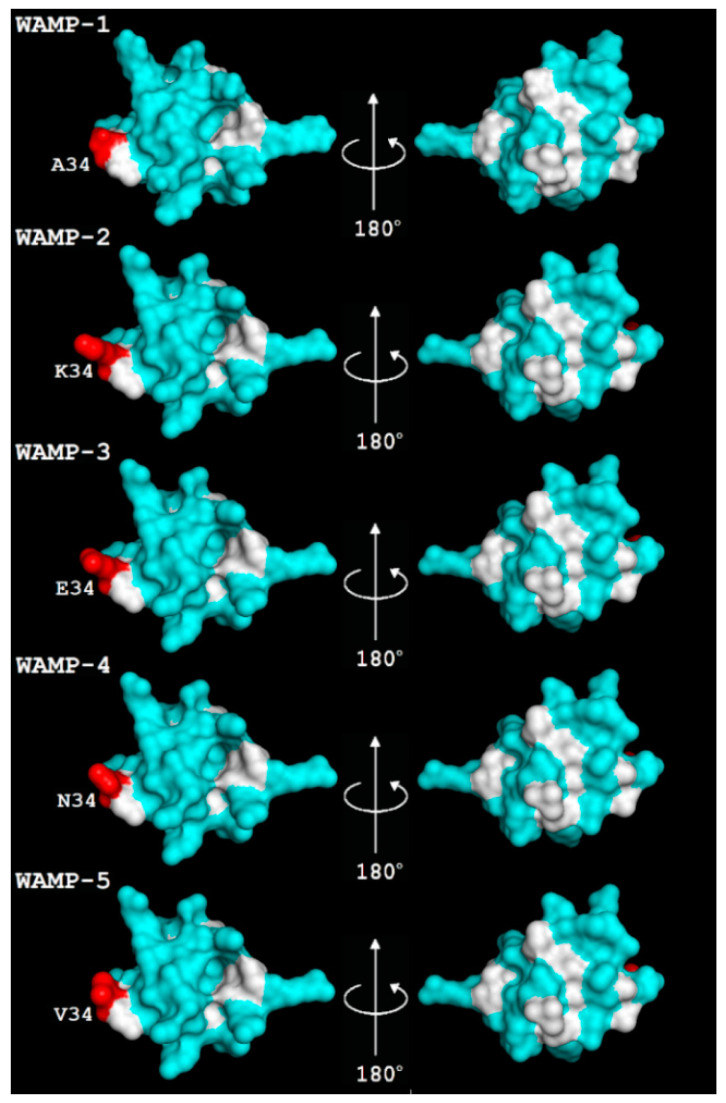
The spatial structure of WAMP-1a (PDB 2LB7) and 3D structure models of WAMP homologues. Modeling was carried out using SWISS-MODEL [37]. The residue at position 34 is shown in red. Non-polar residues are colored white, and polar residues are colored cyan.

Antifungal activity against *F. oxysporum* decreased as follows: WAMP-3.1˃WAMP-2>WAMP-4˃WAMP-5>WAMP-1b. The efficiency of inhibition of *F. culmorum* spore germination was tested only with WAMP-2, and it appeared to be rather low (IC_50_ = 256.3 µg/mL). All tested peptides inhibited spore germination in *A. alternata*; however, the degree of inhibition was lower than that in *F. oxysporum* and *B. sorokiniana*. It is noteworthy that *C. cucumerinum* was effectively suppressed by WAMP-2. WAMP-4 and WAMP-5 were also effective inhibitors of spore germination of this fungus; their activity did not differ significantly. However, the fungus was insensitive to WAMP-1b and WAMP-3.1 at the concentrations tested.

### 2.3. WAMP-2-Derived Peptides

#### 2.3.1. Design

The amino acid sequences of WAMP-2-derived peptides are shown in Figure 3. The peptides WAMP-N and WAMP-C correspond to the N-and C-terminal regions of the molecule, respectively, while WAMP-G1 and WAMP-G2 correspond to its central part. The sequences of WAMP-G1 and WAMP-G2 were selected as follows. In WAMP-2, we identified two overlapping regions that corresponded to the gamma-core defined by Yount and Yeaman [38] as the multidimensional signature with the GXCX_3-9_C or CX_3-9_CXG motif. The gamma-core is postulated to be vital for the antimicrobial activity of AMPs [38,39]. The sequence of WAMP-G2 included the gamma-core **G**F**C**GSGDAY**C** and four amino acid residues located N-terminally to the gamma-core. It also harbored all three aromatic residues of the chitin-binding site involved in carbohydrate binding (Tyr22, Phe24 and Tyr31 in WAMP-2). WAMP-G2 corresponded to beta2, beta3, the alpha-helix and their connecting loops in WAMP-2 three-dimensional structure (Figure 3). These secondary structure elements constitute the central structural domain conserved in all plant hevein-like AMPs. WAMP-G1 includes the second putative gamma-core **C**GKY**G**F**C** in a levomeric form and two and three additional amino acid residues from the N- and C-termini, respectively. WAMP-G1 is shorter than WAMP-G2 by four amino acid residues from the C-terminus and lacks the last Tyr residue of the chitin-binding site. WAMP-G1 includes two beta-strands (beta2 and beta3) and the interposed turn region in the WAMP-2 three-dimensional structure (Figure 3). This multidimensional signature is postulated to be a characteristic feature of the gamma-core [38]. In the WAMP-2 molecule, WAMP-N, WAMP-G1/G2 and WAMP-C peptide regions form adjacent clusters on the surface of the molecule (Figure 3).

#### 2.3.2. Physicochemical Characteristics

The calculated physicochemical characteristics of WAMP-2-derived peptides are presented in Table 2. The molecular weights of the peptides are in the range from 1194.40 to 1533.73 Da, and their length ranges from 12 to 15 amino acid residues. The isoelectric point (pI) values vary from 5.81 in WAMP-G2 to 9.22 in WAMP-C. Three peptides, WAMP-C, WAMP-N and WAMP-G1, are positively charged, while WAMP-G2 is not charged at neutral pH. WAMP-C has the highest net positive charge of (+3) due to the presence of two Lys and one Arg residues. WAMP-N also has three positively charged residues; however, its net charge is lower (+2) due to the presence of a negatively charged Glu residue. WAMP-G1 has a single positively charged Lys residue, while in WAMP-G2, this residue is compensated by a negatively charged Glu leading to the lack of charge in WAMP-G2. Most AMPs have a net positive charge of the molecule (at least +2) due to overrepresentation of positively charged residues Lys and/or Arg, and the lack or low number of negatively charged residues Glu and/or Asp; however, anionic AMPs have also been reported [40]. It has been generally acknowledged that a positive charge allows AMPs to interact with negatively charged membranes of the pathogens [41].

The aliphatic index defined as the relative volume of the side chains of Val, Ala, Ile and Leu proposed by Ikai [42] positively correlates with thermal stability of a protein. For thermophilic proteins it makes about 100 [43]. The aliphatic index of WAMP-2-derived peptides is not high and varies from 6.67 to 32.5 in WAMP-G2, WAMP-N and WAMP-G1, and for WAMP-C it is equal to 0 because of the absence of the above-mentioned amino acids in the peptide (Table 2).

The instability index proposed by Guruprasad et al. [44] predicts the stability of a protein based on its amino acid composition. The peptide is considered stable if this index does not exceed 40. For all WAMP-2-derived peptides, except for WAMP-C, this index is below 40 (Table 2).

The hydrophobicity of a peptide is determined by the GRAVY (Grand Average of Hydropathicity) index, which is the measure of the peptide’s solubility [45]. The GRAVY value is defined by summing hydropathy values of all amino acids and dividing by the protein length. Proteins with a positive GRAVY index are more hydrophobic and less soluble, and proteins with negative index are more hydrophilic and dissolve better in water. A low GRAVY index is a characteristic feature of thermostable proteins [43]. Hydrophobicity is the peptide’s property that targets the peptide into the membrane. WAMP-N and WAMP-C are characterized by negative GRAVY indices, while WAMP-G1 and WAMP-G2 are likely to be more hydrophobic (Table 2).

The Boman index [46], originally called the protein-binding potential, is estimated as the sum of the solubility values for all residues in a sequence divided by the number of residues. It reflects the affinity of a peptide to proteins and its ability to participate in biological interactions. A protein has high binding potential if this index is higher than 2.48. A more hydrophobic peptide tends to have a negative index, while a more hydrophilic peptide tends to have a more positive index [46]. This index allows one to differentiate between the protein–protein (hormones) or protein–membrane (AMPs) interactions. For AMPs, the Boman index is usually negative or close to 0. The Boman index values for WAMP-2-derived peptides range from 0.53 to 3.58 kcal/mol (Table 2). WAMP-G1 has a negative Boman index characteristic of hydrophobic peptides. WAMP-G2 has the Boman index of 0.28. Thus, both WAMP-G peptides are likely to interact with membranes. For WAMP-N and WAMP-C, this index exceeds 2.48 kcal/mol, suggesting their high protein-binding potential.

#### 2.3.3. Three-Dimensional Structures of WAMP-2-Derived Peptides

The spatial structure of WAMP-2 peptide fragments was predicted using the PEP-FOLD 3 program [47] (Figure 4). Both N- and C-terminal peptides, WAMP-N and WAMP-C, respectively, are predicted to have an alpha-helical region starting from D6 in WAMP-N and C5 in WAMP-C. WAMP-G1 is predicted to have two antiparallel beta-strands. The WAMP-G2 is likely to adopt a random coil confirmation; however, a single disulfide bond between the cysteine residues C2 and C15 is predicted. The distribution of polar and non-polar residues in all peptides is uneven, which is typical for AMPs. WAMP-N and WAMP-C are predominantly polar. The surface of WAMP-C is virtually entirely hydrophilic *(*Figure 4). The surface of WAMP-N peptide is also mostly hydrophilic; however, it has a small hydrophobic cluster. WAMP-G1 has a central hydrophobic core surrounded by hydrophilic side chains, which do not form a separate cluster. WAMP-G2, the longest peptide among all WAMP-2-derived peptides, has a polar face and a smaller hydrophobic cluster (Figure 4).

#### 2.3.4. The Antifungal Activity of WAMP-2-Derived Peptides

The antifungal activity of WAMP-2-derived peptides was assayed against seven plant pathogenic fungi, causative agents of cereal and vegetable diseases, including five pathogens tested with the intact WAMP-2 peptide (Figure 5, Table 3).

The activity of peptide fragments varied depending on the fungus tested. There was also considerable variation in the antifungal activity of different peptides. WAMP-C was active against four fungi: *C. cucumerinum. A. alternata*, *P. nodorum*, and *B. sorokiniana*. The activity against other fungi was low: 23% inhibition of spore germination in *F. culmorum*, and 11.5% inhibition of spore germination in *F. oxysporum* at the highest tested concentration of 400 μg/mL (Figure 5).

The highest activity was observed against *C. cucumerinum*; at a concentration of 50 μg/mL, inhibition of spore germination was 70.2%, and at the maximal tested concentration, it amounted to 91%. The activity against *P. nodorum* reached 59.9% at the highest tested concentration. The WAMP-G1 peptide was the least active. The highest activity was observed against *B. sorokininana*, with 55% inhibition at 400 μg/mL. Against other fungi, WAMP-G1 was inactive (*A. alternata*, *F. avenacenum* and *F. culmorum*), or weakly active: 16.7% inhibition in the case of *F. oxysporum*, 19.2% inhibition of *C. cucumerinum*, and 28.5% inhibition of *Parastagonospora nodorum* spore germination at 400 μg/mL. WAMP-G2 was significantly more active than WAMP-G1. It inhibited spore germination of all tested fungi. The activity decreased as follows: *A. alternata*˃*B. sorokiniana˃F. oxysporum*˃*C. cucumerinum*˃*P. nodorum*˃*F. avenaceum*˃*F. culmorum.* For the most sensitive fungus (*A. alternata*), 83.8% inhibition was achieved at the highest tested concentration (400 μg/mL). At this concentration, inhibition of *F. culmorum* spore germination amounted to only 26.2%. Of all WAMP-2-derived peptides, WAMP-N displayed the highest activity against all fungi, except for *C. cucumerinum*, which was most efficiently inhibited by WAMP-C. The antifungal activity of this peptide decreased as follows: *B. sorokininana*˃*A. alternata*˃*P. nodorum*/*F. oxysporum*˃*C. cucumerinum*˃*F. culmorum*˃*F. avenacenum.* At the highest tested concentration, inhibition of *A. alternata* and *B. sorokiniana* was 95.3 and 92.6%, respectively, while for the least sensitive fungus *F. avenacenum*, it was only 40.4%.

#### 2.3.5. The Synergistic Interaction of WAMP-2-Derived Peptide Fragments with Tebuconazole

The sensitizing potential of the WAMP-2 fragments was assayed against five fungi, causing such harmful cereal diseases as foot/root rot and kernel smudge (*Fusarium* spp., *B. sorokiniana* and *A. alternata*), head blight (*F. culmorum*, *F. avenacenum*), and glume blotches (*P. nodorum*).

To assess the ability of fragments to augment the antifungal effect of tebuconazole, the fragments were used at marginally or moderately fungitoxic concentrations, while tebuconazole was co-applied at sub-fungicidal dosages. For WAMP-2-derived peptides, the corresponding concentrations were selected based on their antifungal activity against each cereal pathogen (Figure 5). They ranged from concentrations providing a minimal statistically significant inhibitory effect to concentrations at which suppression of the conidian germination did not exceed 50%. The only exception was the use of WAMP-G1 in non-toxic doses against *F. avenaceum* completely insensitive to this short peptide up to the highest concentration tested (400 μg/mL). The fungicide concentrations, which caused 5–40% germination suppression (except one case for *F. avenaceum*, when the highest inhibition level was almost 50%) were considered as sub-fungicidal ones. The inhibitory effect of all possible combinations of each peptide and the fungicide at concentrations belonging to the indicated ranges was determined by a checkerboard assay [48] and compared with the level of inhibition after spore exposure to the same concentrations of the fungicide or the peptide fragment taken alone. As a result, the tested WAMP-2 fragments were shown to efficiently enhance the inhibitory effect of Folicur^®^ EC 250, an industrial tebuconazole formulation, applied in agriculture to combat the abovementioned cereal pathogens.

Treatments with the fungicide combined with WAMP-2 fragments inhibited fungal spore germination at a much greater level than the fungicide alone, and the type of interactions between the fungicide and the peptide fragment was either synergistic or additive, depending on the target fungus and concentration combinations of the fragments and Folicur^®^.

Among the WAMP-2-derived peptides, the N- and C-terminal fragments most often demonstrated a synergy with the fungicide resulting in complete inhibition of spore germination in fungi. After exposure of fungal conidia to all examined Folicur^®^ combinations with WAMP-N (Figure 6) or WAMP-C (Figure 7), a drastic increase in the sensitivity to tebuconazole was revealed in all tested fungi, including *F. avenacenum* almost insensitive to WAMP-C. Synergistic interactions were confirmed by both a significant exceedance of the real inhibitory effect (Er) over the expected additive effect of the fungicide and the peptide fragment (Ee) [49] and the values of fractional fungicidal concentration indices (FFCIs) [30], which were found to be much lower than 0.5 (Table 4). In some cases, the tebuconazole doses, which, taken alone, displayed only sub-fungicidal antifungal activity, provided an absolute or almost complete suppression of spore germination when used as fractional concentrations in combinations with WAMP-N (Figure 6) or WAMP-C (Figure 7, *BSOR*, *FCUL*, *FAVE*, *PNOD*).

Compared to other cereal pathogens *A. alternata* was less responsive to the sensitization by the C-terminal fragment WAMP-C. While the sensitivity of this causative agent could be synergistically increased by the fungicide in combination with WAMP-C at its fractional concentrations of 100 or 200 μg/mL (Figure 7, *AALT*), the peptide interactions at ≤ 50 μg/mL with the same sub-fungicidal Folicur^®^ dosages (0.005–0.2 μg/mL) were additive.

As in estimating the antifungal activity of WAMP-G1, *B. sorokiniana* was found to be the most responsive among the pathogens to the sensitizing activity of this peptide. The highest synergistic effect was observed when WAMP-G1 at one nominal (50 µg/mL) and two moderately inhibiting (100 and 200 µg/mL) concentrations was combined with non-fungicidal or sub-fungicidal Folicur^®^ dosages (Figure 8). Four of twelve such concentration combinations caused almost complete or absolute suppression of conidian germination. In these cases, Ee values varied from 34.1–48.3% or 52.6–60.8%, depending on the fragment (100 or 200 µg/mL) and the fungicide (0.2 or 0.5 µg/mL) concentrations used, while Er ranged from 96.8 to 100% (Figure 8, *BSOR*), and FFCI was 25 times lower than 0.5 (Table 4), strongly confirming the synergy [30,50,51].

An augmentation of the tebuconazole efficacy through WAMP-G1 co-applications was also observed towards the fungi, which demonstrated nominal *(F. culmorum*), low (*P. nodorum*) or null sensitivity to this fragment (*A. alternata*, *F. avenacenum*), but the type of WAMP-G1–fungicide interaction differed markedly depending on the pathogen. In experiments with *F. avenacenum* and *P. nodorum*, an obvious synergism between the compounds was detected, except for only the combination 50 + 0.05 µg/mL in the case of *P. nodorum* (Figure 8, *FAVE*, *PNOD*). At the same time, most combinations tested against *F. culmorum* provided an additive effect, while only three of them pointed to synergistic interactions (Figure 8, FCUL). In addition, WAMP-G1 at non-toxic concentrations (from 50 to 200 µg/mL) did not sensitize *A. alternata*, another fungus insensitive to this fragment, and insignificantly enhanced the effect of Folicur^®^ against this pathogen in an additive manner at a fractional concentration of 400 µg/mL (7.2% inhibitory effect, alone). In the latter case, WAMP-G1 co-applications with the fungicide doses from 0.01 to 0.1 µg/mL resulted in Ee varied from 13.0 to 39.6%, and Er varied from 14.4 to 42.2%.

Compared to WAMP-G1, WAMP-G2 was shown to be a more active sensitizing fragment, synergistically exacerbating the effect of sub-fungicidal Folicur^®^ dosages against all cereal pathogens including *A. alternata* (Figure 9, Table 4), although two (100 + 0.05 and 100 + 0.1 µg/mL) of nine compositions assayed toward this fungus gave an additive effect. In several cases, the antifungal efficacy of tebuconazole against *B. sorokiniana*, *F. avenacenum* and *P. nodorum* could be augmented to the level exceeding 90% due to co-application with WAMP-G2 (Figure 9).

## 3. Discussion

Hevein-like AMPs represent a group of structurally related cysteine- and glycine-rich peptides found in plants of different families [11]. They show similarity to hevein and chitin-binding domains of chitin-binding proteins, such as lectins, chitinases and wound-inducible proteins, because they possess a chitin-binding site. The chitin-binding site composed of several conserved amino acid residues, among which there are three aromatic residues and one serine, determines the ability of hevein-like AMPs and other chitin-binding proteins to bind chitin, a polymer of N-acetylglucosamine units linked by beta-(1-4)-glycosidic bonds, which is the main constituent of fungal cell walls. The cysteine residues forming disulphide bridges provide structural stability to the peptide molecule. Hevein-like AMPs differ in the number of cysteine residues and are subsequently grouped into three subfamilies: 6-Cys-, 8-Cys- and 10-Cys-containing peptides. 6-Cys-containing hevein-like AMPs are believed to be truncated variants of hevein and other 8-Cys-containing peptides, since they lack the disulphide bond between the two cysteines in the C-terminal region of the molecule. The subfamily of 10-Cys peptides is the most structurally diverse: the position of the 5th disulphide bond differs in different subfamily members. Three disulphide bridges in hevein-like peptides are strictly conserved. Of them, two adjacent disulphide bonds (C1-C4, C2-C5) are located perpendicular to each other and are involved in the formation of a knottin-like core. The overall fold of hevein-like AMPs consists of a central beta-sheet, comprising at least two antiparallel beta-strands, and one or two short helical regions. The structure of the chitin-binding motif composed of two antiparallel beta-strands and a helical region, is well conserved among hevein-like AMPs.

The mode of action of the hevein-like AMPs remains poorly studied. However, their antifungal activity is supposed to be related to their chitin-binding activity. Thus, for the hevein-like peptide Cy-AMP1 from cycad seeds, an essential role of the chitin-binding capacity in antifungal, but not antimicrobial, activity was shown [13]. The role of the chitin-binding domain in antifungal activity was studied for other chitin-binding proteins. Chitinases containing the chitin-binding domain are usually more active against fungi than those without it [52,53,54]. The fusion of a chitin-binding domain of the rice chitinase to an antibacterial defensin alfAFP enhanced resistance to the fungus *Fusarium solani* in transgenic tobacco (*Nicotiana tabacum* L.) [55].

It was speculated that interactions of hevein-like AMPs with chitin interefere with cell wall morphogenesis in fungi. Cell walls of most fungi consist of microfibrils of chitin and beta-1,3-glucans in a matrix of polysaccharides (alpha-1,3-glucans) and glycoproteins (mainly galactomannoproteins) [56]. During cell wall expansion, chitin is synthesized by chitin synthases localized in membranes. The interference of hevein-like AMPs in chitin biosynthesis was first suggested for hevein [57]. Hevein is localized in the vacuole-derived lutoid bodies of *H. brasiliensis*. It was suggested that upon pathogen attack, hevein is released into the cytoplasm together with an array of lytic enzymes. Due to its small size and affinity to chitin, the peptide penetrates through the fungal cell walls and interferes with fungal growth by binding or cross-linking the newly synthesized chitin chains, thus violating the “steady-state” model of hyphal growth proposed by Wessels [58], which postulates that the apical cell wall is expandable and becomes rigid by the action of cross-linking enzymes. According to an alternative unitary fungal growth hypothesis, there exists a balance between the chitin synthesis and the hydrolysis of preformed chitin chains realized through a coordinated action of cell wall-loosening and cell wall synthesizing enzymes [59,60], which might be disturbed by hevein.

Another hypothesis explaining the fungistatic effect of hevein-like AMPs was put forward by Koo et al. [61]. Analyzing the mode of action of the 8-Cys-containing hevein-like peptides Pn-AMPs from *Pharbitis nil* (L.) Choisy, the authors revealed similarity with thionins. Using confocal microscopy, they demonstrated that Pn-AMPs rapidly penetrated into the fungal hyphae and localized at septum and hyphal tips, resulting in hyphal tip burst and leakage of the cytoplasmic constituents.

The anionic 6-Cys-containing hevein-like peptide roseltide rT7 from *Hibiscus sabdariffa* L. (Malvaceae family) was also shown to have intracellular targets [62]. Confocal microscopy studies demonstrated that after roseltide rT7 penetrates cells, it causes accumulation of ubiquitinated proteins and inhibits human 20S proteasomes. It was also shown that the IIML motif located in the proline-rich loop 4 of roseltide rT7 is involved in proteasomal inhibition.

For the 10-Cys-containing hevein-like peptide Ee-CBP from the spindle tree (*Euonymus europaeus* L.), using the same approach (confocal microscopy) Van den Bergh et al. [63] discovered that Ee-CBP binds to the surface of the fungal spores and hyphae. In the case of *B. cinerea*, which is the most sensitive fungus to the Ee-CBP peptide, the peptide was also found inside fungal cells. However, contrary to the results obtained with the Pn-AMP peptides, Ee-CBP was not discovered in the septa of the moulds tested. The Ee-CBP peptide also bound to the surface of *Pythium ultimum*, the cell walls of which are devoid of chitin, suggesting that the peptide can interact not only with GlcNAc-oligomers and chitin, but with other cell wall components as well.

In our previous studies, we isolated and structurally characterized WAMPs, 10-Cys-containing hevein-like peptides from *T. kiharae*, which display broad-spectrum antifungal and antibacterial activities [15]. Furthermore, we revealed structural similarity between WAMPs and the chitin-binding domains of the class 1 chitinases from cereals that are subject to proteolytic degradation by fungal effectors, secreted Zn-dependent metalloproteinases termed fungalysins. We further demonstrated that in contrast to plant chitinases, WAMPs are not cleaved by fungalysin but conversely act as effective fungalysin inhibitors, and the residue at position 34 affects the efficiency of inhibition [20]. WAMP homologues with Ala and Lys at this position effectively inhibit fungalysin, while those with Glu and Asn, do not; however, they still retain the ability to suppress spore germination of the plant pathogenic fungi in vitro. Recent studies of the secreted fungalysin metalloprotease of *Ustilago maydis* showed that the enzyme has a dual function in the fungal morphogenesis and plant pathogenesis by modulating activity of both fungal and plant chitinases [64]. Thus, we believe that the inhibition of fungalysin by WAMPs can play a plant protective role in two ways: (i) by rescuing plant defense chitinases from proteolysis by fungalysin and thus restoring their ability to degrade fungal cell walls, (ii) by protecting fungal chitinases involved in fungal cell wall remodeling during the morphogenesis. However, the question remained whether the inhibition of fungalysin is the only mechanism employed by WAMPs to suppress the growth of fungi and other pathogens. The hypothesis on the occurrence of a fungalysin-independent mode of action arose from the observation that WAMPs that did not inhibit fungalysin still preserved the ability to inhibit *Fusarium* and other fungi [20]. A possibility still existed that, in addition to fungalysin inhibition, WAMPs could interfere with chitin synthesis by binding chitin oligomers through its chitin-binding site. However, interactions of WAMPs with carbohydrates were weaker than those of hevein and other hevein-like AMPs due a serine to glycine substitution in the chitin-binding site. The serine residue forms a hydrogen bond with the acetomide moiety of a GlcNAc residue that stabilizes the complex between the aromatic residues and chitin. WAMP-1a failed to bind penta-*N-*acetylchitopentaose in vitro [16], although the binding to polymeric chitin was preserved [15]. Suppression of chitin synthesis as an alternative to fungalysin-mediated mode of action still could not explain the ability of WAMPs to inhibit the growth of pathogens devoid of chitin. Despite the fact that the serine/glycine replacement in the chitin-binding site mitigates the carbohydrate binding capacity of WAMPs, it increases the peptides’ amphiphilicity [16], which is an important characteristic of membrane-active AMPs. A membrane-disruptive mechanism could thus be suggested as another mechanism of action of WAMPs. In summary, all of the above considerations support the existence of more than one mechanism of action for WAMPs.

In our study, to further explore the antifungal mechanism of WAMPs, we carried out a systematic analysis of the antifungal activity of WAMP homologues against a panel of five fungi and studied the effect of synthetic peptides derived from different regions of the WAMP-2 molecule on the spore germination of important plant pathogenic fungi (Figure 5). We showed that WAMPs that differ by a single amino acid residue essential for the efficiency of fungalysin inhibition differ in their antifungal potency against the five fungal pathogens tested. The antifungal activity of the peptides depends on the fungus and does not correlate with the degree of fungalysin inhibition. As we mentioned above, WAMP-1b and WAMP-2 inhibited fungalysin from *F. verticillioides*, while WAMP-3.1 and WAMP-4 did not do this [20], although all WAMPs exhibited antifungal activity in vitro. Thus, WAMP-3.1 (non-inhibitor of fungalysin) was the most active peptide against *F. oxysporum*, while WAMP-1b (fungalysin inhibitor) was the least active. WAMP-2 (fungalysin inhibitor) was the least active against *B. sorokiniana*. The efficiency of inhibition of *A. alternata* spore germination was the highest for WAMP-5, followed by WAMP-1b/WAMP-3.1. WAMP-2 and WAMP-4 were the least active against this fungus. These results suggest that in addition to fungalysin inhibition, there exists a fungalysin-independent mechanism for the antifungal action of WAMPs. The observed variation in the antifungal activity may be related to different physicochemical properties of the residues at position 34 in WAMPs; Ala and Val are hydrophobic non-polar residues, while Lys, Asn and Glu are polar and hydrophilic. Lys is basic and Glu is acidic. The size of the side chains is also different and decreases as follows: Lys ˃ Val/Glu ˃ Asn ˃ Ala. Therefore, the introduced mutations changed the properties of the entire WAMP molecules. The net charge in WAMP homologues at pH 7 varies from +3 to +5, and pI values vary from 8.41 to 8.78.

To further explore the role of different parts of the WAMP molecule in antifungal activity, short peptides corresponding to the N-terminal, central and C-terminal regions of the WAMP-2 molecule were synthesized, and their effect on spore germination of seven fungi was assayed.

The results showed that the short WAMP-2-derived peptides differed in antifungal activity against the tested fungi. Similarly to intact WAMPs, there was also considerable variation in antifungal potency of a given peptide against different fungi. A remarkable fact to be noted is that WAMP-C was extremely highly and selectively active against *C. cucumerinum*, much more active than the whole WAMP-2 (Table 3). This finding indicates that the activity of WAMP-2 against *C. cucumerinum* is not associated with the chitin-binding site but depends on the C-terminal region of the molecule. WAMP-C has the highest positive charge (+3) of all WAMP-2-derived peptides, suggesting electrostatic interactions with negatively charged groups in fungal cell walls and/or membranes. The peptide contains a predicted helical region, characteristic of many alpha-helical AMPs. However, the surface of WAMP-C is predicted to be mostly hydrophilic, indicating that the formation of pores and penetration through membranes as a mechanism of the peptide’s antifungal action is unlikely. Of all WAMP-2-derived peptides, WAMP-N exhibited the highest activity against all fungi, except for *C. cucumerinum*, pointing to the importance of the N-terminal region of the WAMP-2 molecule for antifungal activity. The physicochemical properties of WAMP-N are similar to that of WAMP-C, although it is less positively charged (net positive charge of +2) and has a lower pI of 8.96 versus 9.22 for WAMP-C, and it is more stable. The WAMP-N is predicted to be alpha-helical and amphiphilic, suggesting the penetration through fungal membranes as a possible mechanism of action.

Studies of the antifungal potency of the two closely related overlapping peptides WAMP-G1 and WAMP-G2 derived from the central part of the WAMP-2 molecule and involving the putative gamma-core regions revealed striking differences in their antifungal potency. The longer peptide WAMP-G2, comprising, in addition to the gamma-core **G**F**C**GSGDAY**C**, all three aromatic residues of the chitin-binding site, was much more active than WAMP-G1, which was the least active of all WAMP-2-derived peptides. For all fungi except *B. sorokiniana*, IC_50_ values for WAMP-G1 exceeded 700 µg/mL (Table 3). Please note that WAMP-G1 contains a putative gamma-core **C**GKY**G**F**C** and only two conserved aromatic residues of the chitin-binding site (Tyr22 and Phe24). The peptide contains two antiparallel beta-strands with an interposed turn region, postulated to be a characteristic feature of the gamma-core multidimensional signature [38]. Given that WAMP-G2 differed from WAMP-G1 by four amino acid residues from the C-terminus, including Tyr31, our finding that WAMP-G2 was much more active than WAMP-G1 allows us to speculate that all three conserved aromatic residues of the chitin-binding site contribute to the antifungal activity of WAMP-G2. WAMP-G2 was found to be active against all fungi except *F. culmorum*, for which the IC_50_ value was above 700 µg/mL. However, the antifungal activity of WAMP-G2 was lower than that of WAMP-N, suggesting that the N-terminal region of the WAMP-2 molecule plays a key role in antifungal activity against most tested fungi (except for *C. cucumerinum*, which was most efficiently inhibited by WAMP-C). Thus, the contribution to the antifungal activity of WAMPs of other peptide regions beyond the chitin-binding site, mainly the N-terminal region, becomes evident.

We also evaluated the sensitizing potential of the WAMP-2-derived peptides added to tebuconazole, the active ingredient of a triazole fungicide Folicur^®^ EC 250, using the fragments taken at concentrations that had marginal or moderate impact on the spore germination. Independently of the individual antifungal activity of WAMP-2-derived peptides, all of them, including the least toxic G1, significantly enhanced the pathogen sensitivity to tebuconazole when were combined with this fungicide.

The fungicidal compositions involving N- or C-terminal fragments provided synergistic activity against all tested cereal-damaging pathogens. It suggests that these WAMP-2 terminal fragments can be used to augment the Folicur^®^ efficacy against both soil-born (*Fusarium* spp., *B. sorokiniana*, *A. alternata*) and foliar (*P. nodorum*, *B. sorokiniana*) pathogens often forming common pathogenic complexes on cereals [65,66,67,68]. Based on the results of the microplate assay, several WAMP-2 fragment–fungicide compositions, which inhibited the spore germination in all tested fungi at the level ≥ 90%, were found (e.g., Folicur^®^ at 0.2 µg/mL and WAMP-C at 200 µg/mL on Figure 7) and thus seem to be promising for in planta testing against multiple wheat pathogens.

Some cyclic lipopeptides (CLPs), representing bacterial AMPs produced by non-ribosomal biosynthesis, which had no inhibitory activity against the spore germination at low concentrations, were reported to show a significant synergistic inhibitory effect on the germination of *F. graminearum* conidia in combinations with one another [69]. Co-application of the fermentation broth of *Bacillus amyloliquefaciens* JCK-12 producing these CLPs with triazoles, benzimidazoles or phenylpyrroles was accompanied by a synergistic in vitro antifungal effect and significantly increased the efficacy of the pathogen control under greenhouse and field conditions. Among plant defense AMPs, the ability to synergistically interact with fungicides resulting in reduced biofilm formation in *Candida albicans* was shown for radish defensins [70]. The ability of short peptides derived from the hevein-like peptides WAMPs to exacerbate the sensitivity of plant pathogenic fungi to a commercial antifungal agent is, to our knowledge, a novel and previously uninvestigated property discovered in our studies and first reported here. It cannot be excluded that the discovered sensitizing activity of WAMP-2 fragments, as with JCK-12 CLPs [69], is associated with the disturbance of the cell membrane permeability.

## 4. Materials and Methods

### 4.1. Plant Pathogenic Fungi

Wheat-pathogenic *Fusarium culmorum* OR-02-37, *F. avenaceum* Br-04-60, *Alternaria alternata* MRD1-12, and barley-damaging *Bipolaris sorokiniana* KrD-81 were obtained from the State Collection of Plant Pathogenic Microorganisms at the All-Russian Research Institute of Phytopathology (ARRIP), while cucumber-damaging *Cladosporium cucumerinum* C5, and wheat kernels artificially infested with *Parastagonospora nodorum* B-9/47 were kindly supplied by the ARRIP Departments of Molecular Biology, and Mycology, respectively, and *F. oxysporum* F37 pathogenic for tomato was from the working collection of the ARRIP Laboratory of Physiological Plant Pathology.

### 4.2. Recombinant Production of WAMPs

Recombinant WAMP analogues with substitutions at position 34 were obtained using site-specific mutagenesis as fusion proteins with thioredoxin, as described earlier [20,35]. The synthetic genes encoding WAMPs were produced by PCR using oligonucleotide primers designed on the basis of the WAMP amino acid sequences. The primers used have been reported previously [19,20]. The target fragment was amplified with a forward primer containing the *Kpn*I restriction endonuclease recognition site and a Met codon for cyan bromide (CNBr) cleavage and a reverse primer containing the BamHI restriction endonuclease recognition site and a stop codon. PCR fragments encoding mature polypeptides were purified using a DNA purification kit, hydrolyzed with restriction endonucleases, and cloned into the pET-32b (+) expression vector, which was also hydrolyzed with the same restriction endonucleases. The correspondence of the obtained structures to the given ones was checked by sequencing. *E. coli* BL21 (DE3) cells were transformed by the resulting constructs, as described earlier [35]. Protein expression was induced by adding IPTG to a concentration of 0.2 mM. The cells were cultured at 25 °C for 12–14 h, harvested by centrifugation, resuspended in the starting buffer for affinity chromatography (20 mm Tris-HCl, 300 mM NaCl, pH 7.5) and destroyed by ultrasound. The fusion proteins were isolated by affinity chromatography on TALON Superflow resin. The hybrid proteins were cleaved with cyan bromide overnight at room temperature in the dark and purified by HPLC on a Luna C8 column (4.6 x 150 mm, Phenomenex) in a linear gradient of acetonitrile (5–25% for 30 min, then 25–60% for 10 min) in 0.1% trifluoroacetic acid. The purity of the target peptides was confirmed by MALDI-time-of-flight mass spectrometry and N-terminal Edman sequencing. The yield of peptides was 0.5–3 mg/L of bacterial culture.

As a result, five WAMP homologues were obtained—WAMP-1b (Ala34), WAMP-2 (Lys34), WAMP-3.1 (Glu34), WAMP-4 (Asn34), and WAMP-5 (Val34)—for the study of their structure–function relationship and antifungal activity.

### 4.3. Chemical Synthesis of WAMP-2-Derived Peptides

Solid-phase chemical synthesis using Fmoc chemistry was used to produce peptides corresponding to particular regions of the WAMP-2 molecule: WAMP-N: AQRCGDQARGAKC, WAMP-G1: LCCGKYGFCGSG, WAMP-G2: CCGKYGFCGSGDAYC, and WAMP-C: GKGSCQSQCRGCR (Cloud-Clone Corp., China). The purification of peptides was performed by RP-HPLC. The identity of the synthesized peptides was verified by mass spectrometry. The purity of peptides was ˃95%.

The following characteristics of the WAMP-2-derived peptides were calculated using ExPASy ProtParam tool [71]: molecular weight, pI, aliphatic index, instability index, GRAVY index, net charge at pH 7. Boman (potential protein interaction) index was computed using APD3 [72].

### 4.4. 3D Structure Modeling

The 3D structure of the WAMPs with substitutions at position 34 was modeled by SWISS-MODEL with WAMP-1a (PDB 2LB7) used as a template [37]. The spatial structure of WAMP-derived short peptides was de novo modeled using PEP-FOLD 3 [47], the best representative models were chosen based on the lowest sOPEP values provided by PEP-FOLD 3.

### 4.5. MALDI-TOF MS

The molecular masses of the peptides were determined by MALDI-TOF MS on an Ultraflex MALDI-TOF mass spectrometer (Bruker Daltonics, Bremen, Germany). Molecular masses were determined in linear or reflector positive-ion mode using alpha-cyano-4-hydroxycinnamic acid as a matrix.

### 4.6. Antifungal Activity Assay

The antifungal effect of WAMP homologues, WAMP-2-derived peptides and tebuconazole formulated as Folicur^®^ EC 250 (a.i. 25% tebuconazole, Bayer AG, Leverkusen, Germany) was evaluated in vitro using a microplate assay by the inhibition of the germination of fungal spores, which were incubated in a 96-well suspension culture plates [73,74]. Conidia of *Fusarium* spp., *A. alternata*, *B. sorokiniana*, and *C. cucumerinum* were collected from the surface of their colonies on potato dextrose agar by flooding the mycelium with sterilized distilled water (SDW) and gently rubbing with a glass rod, while *P. nodorum* spores were obtained by washing off wheat kernels infested with the pathogen. The spores (conidia) were suspended in SDW (control) or in SDW-dissolved WAMPs, WAMP-2-derived peptides, the fungicide or their mixtures to a final concentration of 10^4^–10^5^ spores/mL and incubated at 20–22 °C for 5–6 or 12–16 h depending on the pathogen species. The number of germinated and non-germinated spores (500 ones of each pathogen per treatment) was counted in wells of a polystyrene culture plate using an inverted microscope followed by calculation of the percent of inhibition of the spore germination as compared to control.

### 4.7. Revealing the Synergistic Interactions between WAMP-2-Derived Peptides and Tebuconazole

To assess the ability of chemically synthesized WAMP-2 fragments to enhance the fungicidal effect and examine the levels of spore germination inhibition, the sensitization experiments were designed according to a microdilution checkerboard assay [48]. Fungal spores (conidia) of each target pathogen were exposed to different concentration combinations of Folicur^®^ and each WAMP-2-derived peptide in the suspension culture plates (see Section 4.6) in parallel with exposure of conidia to the same concentrations of these antifungal agents applied separately. Since native WAMP-2 did not show the ability to enhance the inhibitory effect of Folicur^®^ against the wheat pathogens tested, it was excluded from further studies.

The fragments were tested at concentrations ranging between 5, 10 and 50 μg/mL against *B. sorokiniana*, *A. alternata* or other cereal-pathogenic fungi *(F. culmorum*, *F*. *avenacenum* and *P*. *nodorum*), respectively, to concentrations that produced the inhibitory effect of ≤50%. The sub-fungicidal concentrations of tebuconazole were preliminarily determined for each pathogen through serial double or fivefold dilutions.

In the cases where the WAMP-2 fragments showed a detectable germination-suppressing activity (at least 1.5–2.0%), the synergy between a putative sensitizer and the fungicide was confirmed by the Limpel criterion [49] that was determined by the Formula (1):Ee = (X + Y) − XY/100 < Er (at *p* ≤ 0.05)(1)
where Ee is the level of the expected additive effect from the use of two substances (%), X and Y are the percentages of inhibition of the spore germination by each of the substances separately, Er is the percentage of inhibition obtained when a peptide fragment and Folicur^®^ are used together. In addition, fractional fungicidal concentration indices (FFCIs) [30] were determined by Formula (2), which is similar to that commonly used for calculation of fractional growth inhibitory concentration indices [30,50], and, according to generally accepted protocols, FFCIs ≤ 0.5 were interpreted as an evidence of synergistic interactions [50,51].
(2)$$\mathrm{FFCI}=\frac{\begin{array}{c} \mathrm{MFC}\;\mathrm{of}\;a\;\mathrm{WAMP}2\;\mathrm{fragment}\;\\ \mathrm{in}\;\mathrm{combination}\;\mathrm{with}\;\mathrm{the}\;\mathrm{fungicide} \end{array}}{\mathrm{MFC}\;\mathrm{of}\;a\;\mathrm{WAMP}2\;\mathrm{fragment},\;\mathrm{alone}}\;+\frac{\begin{array}{c} \mathrm{MFC}\;\mathrm{of}\;\mathrm{the}\;\mathrm{fungicide}\;\\ \mathrm{in}\;\mathrm{combination}\;\mathrm{with}\;a\;\mathrm{WAMP}2\;\mathrm{fragment} \end{array}}{\mathrm{MFC}\;\mathrm{of}\;\mathrm{the}\;\mathrm{fungicide},\;\mathrm{alone}}$$

The lowest concentrations of WAMP-2 fragments or the fungicide, when used alone, resulting in ≥99.9% inhibition of spore germination were determined by probit-analysis with the involvement of a linear regression and considered as the minimum fungicidal concentrations (MFC). Depending on the pathogen, approximation confidence values (R^2^) varied in MFC calculations from 0.912 to 0.991 or from 0.948 to 0.991 for WAMP-2 fragments or the fungicide, respectively.

### 4.8. Statistical Analysis

For each pathogen, tests on evaluation of antifungal or sensitizing activity of WAMP homologues or their peptides (three replications per treatment, 500 counted conidia in each one) included at least two independent experimental series. Mean values, standard deviations (SD), standard errors (SE), and the significance of differences (*p* ≤ 0.05) of the means between treatments and controls (t-test for the independent variables) were determined using STATISTICA v. 6.1 software (StatSoft Inc., Tulsa, OK, USA).

## 5. Conclusions

The results of antifungal assays with WAMP-2-derived peptides clearly demonstrate that WAMPs exert different modes of action against different fungi. We suppose that inhibition of fungal secreted fungalysin protease that requires the intact hevein domain is not the only WAMP-mediated mechanism of fungal growth suppression. This suggestion follows from our discovery that the N-terminal peptide WAMP-N turned to be the most active against most fungi, while the C-terminal domain appeared exceptionally potent against *C. cucumerinum.* Both peptides are devoid of the chitin-binding site and are predicted to adopt an alpha-helical conformation. The central region of the WAMP-2 peptide comprising all three aromatic residues of the chitin-binding site also contributes to the antifungal activity of WAMPs. The mechanisms underlying the antifungal effect of WAMP-G2 remain unclear. However, we can speculate that binding to chitin nascent chains via the aromatic residues of the chitin-binding site is a possible mechanism. Disturbance of fungal membranes due to the peptide’s amphiphilicity cannot be excluded either. The peptide’s charge seems not to play a key role in the antifungal activity of WAMP-2-derived peptides since WAMP-G2 (net charge 0) was much more active than WAMP-G1 (net charge +1). The gamma-core charge is also not vital (net charge of 0 for WAMP-G2 and +1 for WAMP-G1). As a potential biopesticide, WAMP-C showed the most promise of all the short WAMP-2-derived peptides against *C. cucumerinum*. WAMP-N can be further modified to increase antifungal potency and develop the peptide-based agents against a broad spectrum of important plant pathogenic fungi. Regardless of the structural features and individual antifungal activity, all fragments of the hevein-like peptide WAMP-2 were capable of synergistically enhancing the fungicidal effect of tebuconazole against cereal pathogens. Importantly, some of the fragments may simultaneously serve as effective sensitizers of several fungi. If plant testing confirms our in vitro findings, the availability of these short peptides will be an additional practical advantage, since modern technologies open up the avenue for inexpensive and almost quantitatively unlimited production of oligopeptides including through their direct synthesis. The mechanisms underlying the sensitizing activity of WAMP-2 fragments remain so far unknown; they will be the subject of further investigations.

## Figures and Tables

**Figure 3 ijms-21-07912-f003:**
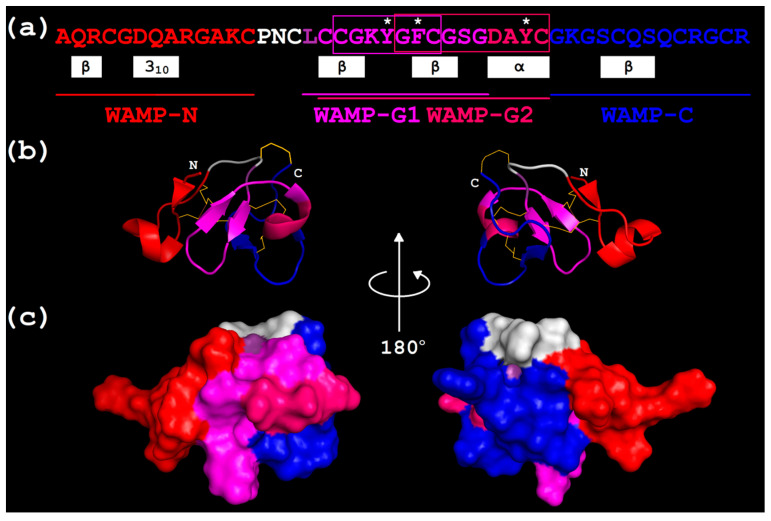
WAMP-2-derived peptides. The position of WAMP-N, WAMP-G1, WAMP-G2, and WAMP-C colored red, magenta, pink and blue, respectively, in the WAMP-2 sequence, and the 3D structure is indicated. (**a**) WAMP-2 amino acid sequence [17]. Secondary structure elements (helices and beta-strands) are shown as white rectangles below the sequence [16]. The conserved aromatic residues of the chitin-binding site are marked with asterisks. The gamma-core regions are framed. (**b**) Superimposition of WAMP-2-derived peptides on the spatial structure of WAMP-1a (PDB 2LB7) (ribbon representation). Beta-strands are shown as arrows and 3_10_ and alpha-helices, as ribbons. The N-and C-termini are indicated by N and C, respectively. Disulphide bridges are shown by thin yellow lines. (**c**) Superimposition of WAMP-2-derived peptides on the surface structure of WAMP-1a.

**Figure 4 ijms-21-07912-f004:**
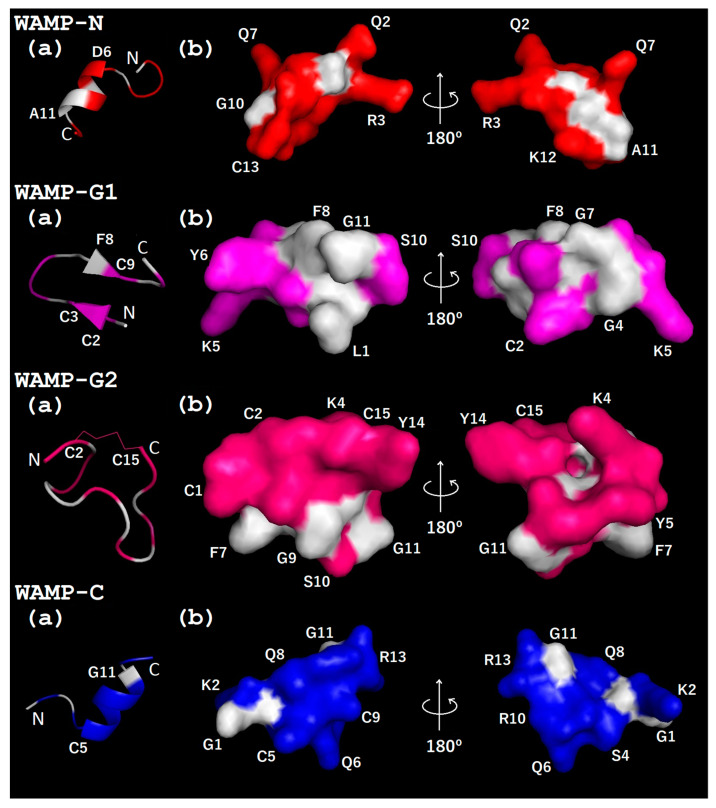
Molecular modeling of the 3D structure of WAMP-2-derived peptides: (**a**) spatial structure (ribbon representation); (**b**) surface structure. A predicted disulphide bond is shown by a thin line. The N- and C-termini are marked with N and C, respectively. Non-polar residues are white, polar residues are colored red, magenta, pink and blue for WAMP-N, WAMP-G1, WAMP-G2 and WAMP-C, respectively. Modeling was carried out using PEP-FOLD 3 [47].

**Figure 5 ijms-21-07912-f005:**
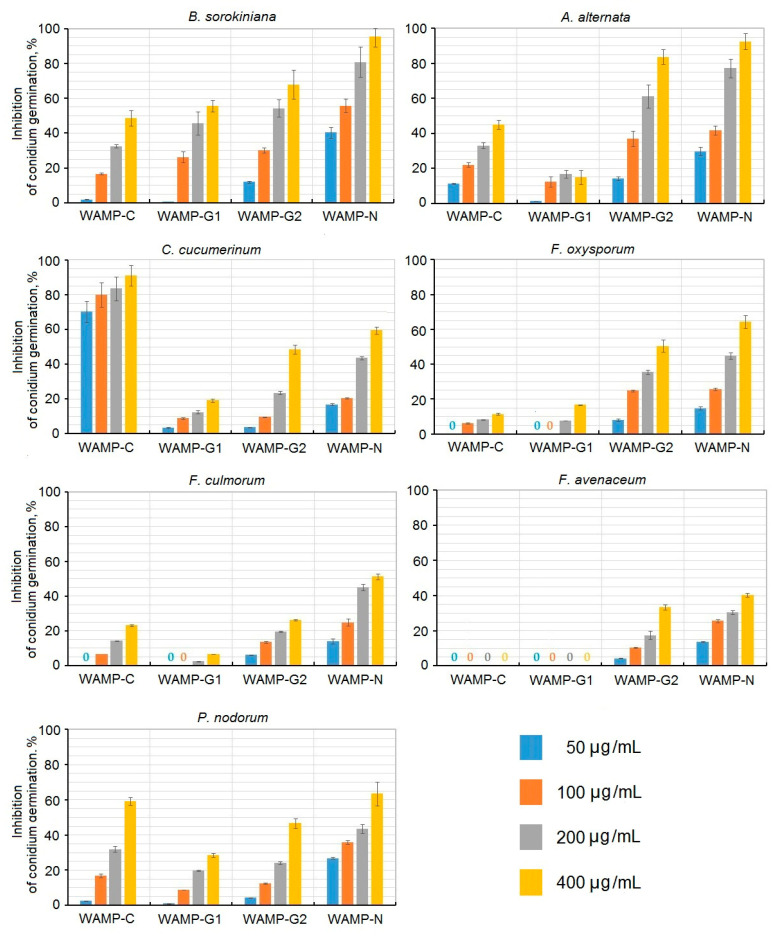
Suppression of spore germination in seven plant pathogenic fungi by WAMP-2-derived peptides. Average means of three experiments for each pathogen and SD are shown.

**Figure 6 ijms-21-07912-f006:**
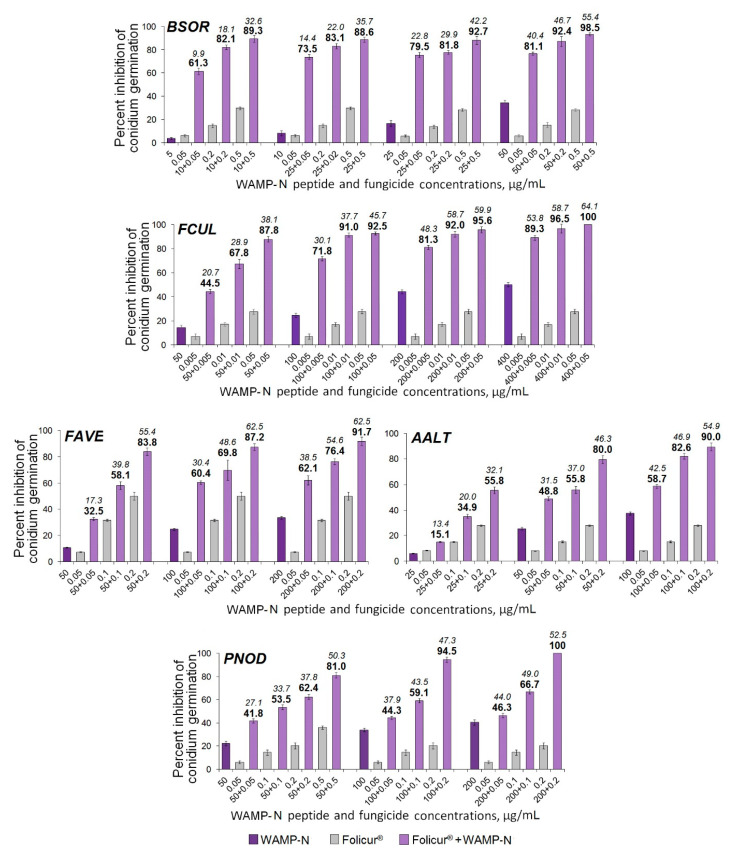
Histograms showing the augmentation of Folicur^®^ EC 250 inhibitory effect on spore germination of five cereal-damaging fungi due to synergistic interactions of WAMP-N with tebuconazole at different concentration combinations of the compounds. *BSOR*, *FCUL*, *FAVE*, *AALT*, and *PNOD* are *B. sorokiniana*, *F.culmorum*, *F. avenaceum*, *A. alternata*, and *P. nodorum*, respectively. The numbers in bold above the columns indicate Er values, while italic numbers show Ee values related to the same fungicide/peptide concentration combination. Er shows percent inhibition of conidian germination when the fungicide and WAMP-N were co-applied; Ee is percent inhibition calculated for an estimated additive effect of the compounds. Er > Ee at *p* ≤ 0.05 points to synergistic interaction of the compounds (see Materials and Methods, 4.7 and [49]). Results are expressed as the mean of two experiments for each pathogen; Y-bars indicate standard error (SE) of the mean (n = 500 conidia in each individual or combined application).

**Figure 7 ijms-21-07912-f007:**
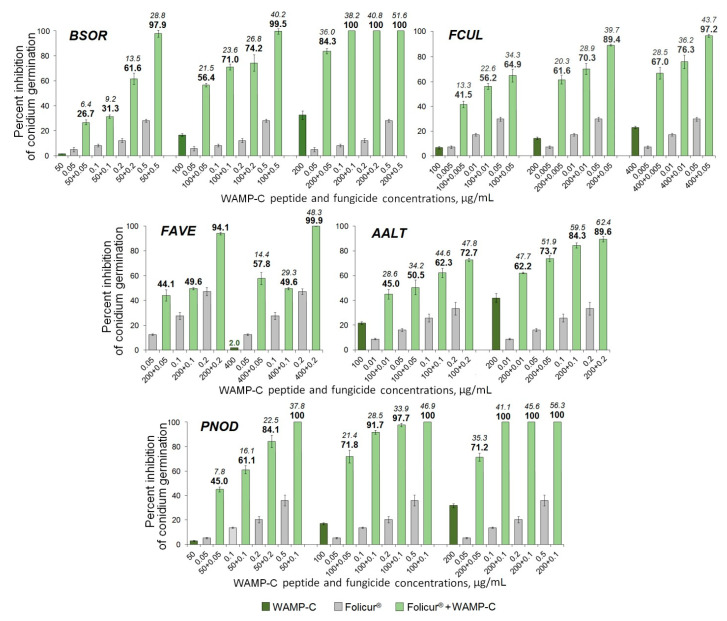
Histograms demonstrating the enhanced Folicur^®^ EC 250 effect on inhibition of spore germination in five cereal pathogens provided by synergy between WAMP-C and the fungicide co-applied in different concentration combinations. *BSOR*, *FCUL*, *FAVE*, *AALT*, and *PNOD*, are *B. sorokiniana*, *F.culmorum*, *F. avenaceum*, *A. alternata*, and *P*. *nodorum*, respectively. Y-bars are SE (n = 500). For additional explanations, see caption to Figure 6.

**Figure 8 ijms-21-07912-f008:**
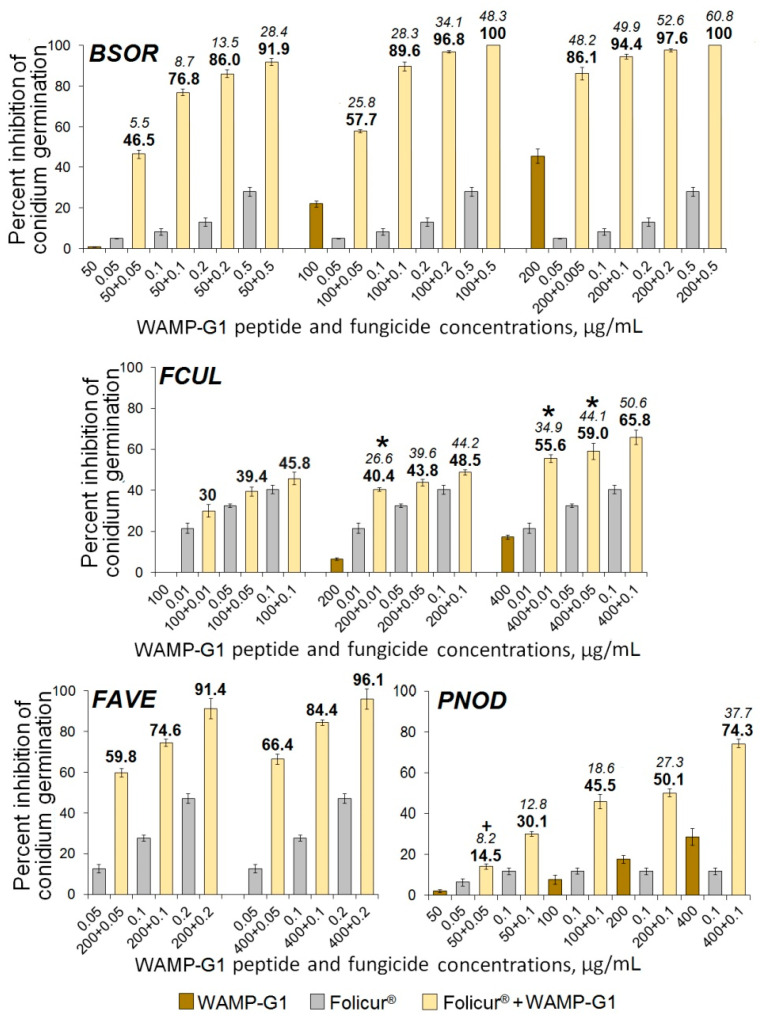
Histograms illustrating Folicur^®^ effect on germination of conidia in four cereal-damaging fungi after the combined application of the fungicide and WAMP-G1. *BSOR*, *FCUL*, *FAVE*, and *PNOD* are *B. sorokiniana*, *F.culmorum*, *F. avenaceum*, and *P*. *nodorum*, respectively. In the case of *FCUL*, the peptide–fungicide concentration combinations resulting in synergistic interaction are marked with asterisks, while the case of an additive effect towards *PNOD* is indicated by plus. For additional explanations, see caption to Figure 6.

**Figure 9 ijms-21-07912-f009:**
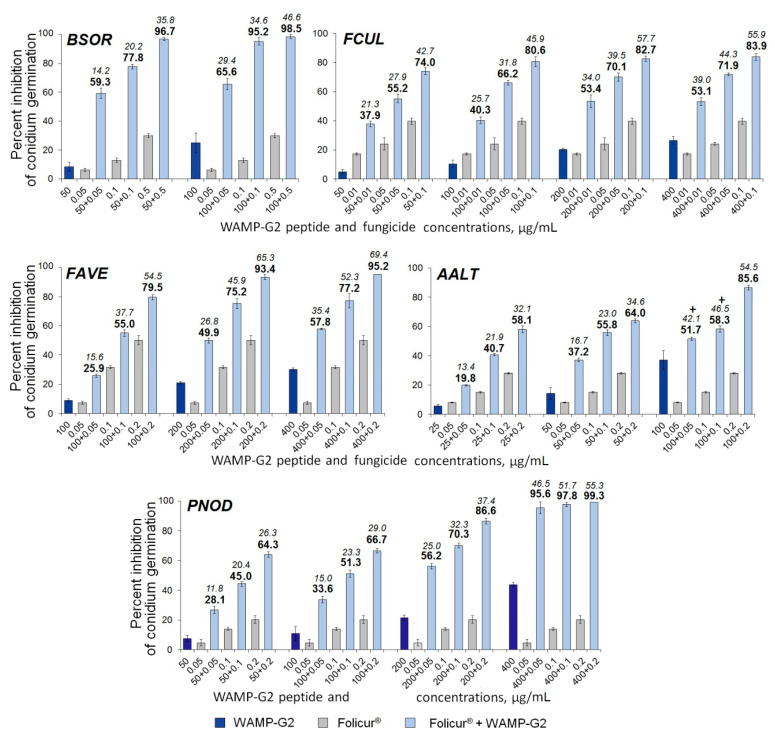
Histograms showing the enhancement of the fungicidal efficacy in suppression of spore germination in five fungi pathogenic to cereal crops due to synergistic or additive interactions between WAMP-N and the fungicide combined at different concentrations. Concentration combinations providing an additive effect are marked with pluses. *BSOR*, *FCUL*, *FAVE*, *AALT*, and *PNOD* are *B. sorokiniana*, *F. culmorum*, *F. avenaceum*, *A. alternata*, and *P*. *nodorum*, respectively. See Figure 6 caption for additional explanations.

**Table 1 ijms-21-07912-t001:** Antifungal activity of recombinant WAMPs.

Fungi	IC_50_, μg/mL				
	WAMP-1b(A34) *	WAMP-2(K34)	WAMP-3.1(E34)	WAMP-4(N34)	WAMP-5(V34) *
*B. sorokiniana*	22.8 ± 3.2	30.6 ± 0,9	22.2 ± 0,5	24.3 ± 1.9	25.2 ± 2.3
*A. alternata*	82.8 ± 7.8	107.3 ± 7.9	82.8 ± 2.8	107.8 ± 13.4	63.8 ± 3.7
*C. cucumerinum*	–	37.1 ± 2.3	–	58.3 ± 3,2	53.5 ± 4.6
*F. oxysporum*	74.8 ± 4.6	41.0 ± 3.2	31.8 ± 1.9	52.3 ± 4.2	56.6 ± 5.1
*F. culmorum*	n/d	256.3 ± 18.1	n/d	n/d	n/d

«–» indicates no activity at peptide concentration below 150 μg/mL; n/d means not determined. * Reported earlier [19]. Numbers in the columns represent the means of two experiments ± SD (see Section 4.8).

**Table 2 ijms-21-07912-t002:** Physicochemical properties of WAMP-2-derived peptides.

Peptide Property	Peptide Name			
	WAMP-N	WAMP-G1	WAMP-G2	WAMP-C
Amino acid sequence	AQRCGDQARGAKC	LCCGKYGFCGSG	CCGKYGFCGSGDAYC	GKGSCQSQCRGCR
Length, aa residues	13 (1–13) *	12 (17–28) *	15 (18–32) *	13 (33–45) *
Molecular weight, Da	1363.53	1194.40	1533.73	1369.55
pI	8.96	7.95	5.81	9.22
Net charge at pH 7	+2	+1	0	+3
Aliphatic index	23.08	32.5	6.67	0
Instability index	−3.83	−4.98	−1.99	51.73
GRAVY index	−1.062	0.542	0.147	−1.169
Boman index, kcal/mol	3.48	-0.53	0.28	3.58

* Denotes position in the WAMP-2 molecule.

**Table 3 ijms-21-07912-t003:** Antifungal activity of WAMP-2-derived peptides.

Fungi	IC_50_, μg/mL			
	WAMP-N	WAMP-G1	WAMP-G2	WAMP-C
*B. sorokiniana*	72.9 ± 12.4	273.1 ± 45.7	195.2 ± 36.8	429.5 ± 51.5
*A. alternata*	102.6 ± 19.6	>700	145.6 ± 29.4	550.4 ± 41.3
*C. cucumerinum*	280.2 ± 29.7	>700	410.2 ± 29.1	5.3 ± 0.1
*F. oxysporum*	238.0 ± 32.4	>700	391.2 ± 34.2	>700
*F. culmorum*	332.4 ± 26.1	>700	>700	>700
*F. avenaceum*	>700	–	602.8 ± 3.1	–
*P. nodorum*	220.2 ± 13.5	>700	424.1 ± 12.1	329.6 ± 7.9

«–» indicates no activity at peptide concentration below 400 μg/mL.

**Table 4 ijms-21-07912-t004:** Synergism-confirming values of Fractional Fungicidal Concentration Indices (FFCIs) in effective tebuconazole–WAMP-2 fragment compositions.

Fungi	FFCIs			
	Sensitizing Peptides			
	WAMP-N	WAMP-G1	WAMP-G2	WAMP-C
*B. sorokiniana*	0.1	0.02	0.13	0.005
*F. culmorum*	0.01	n/d	0.02	0.002
*F. avenaceum*	0.03	n/d	0.03	0.03
*A. alternata*	0.04	n/d, n/s	0.05	0.09
*P. nodorum*	0.06	n/d	0.04	0.09

FFCIs ≤ 0.5 indicate synergistic interactions (see Materials and Methods, 4.7); n/d means not determined for fungi insensitive to WAMP-G1 alone up to the highest tested concentration; n/s means no synergy revealed.

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
