# Peer review of "Hevein-Like Antimicrobial Peptides Wamps: Structure–Function Relationship in Antifungal Activity and Sensitization of Plant Pathogenic Fungi to Tebuconazole by WAMP-2-Derived Peptides"

_ijms, 2020, doi:10.3390/ijms21217912_

Round 1

Reviewer 1 Report

The manuscript by Odintsova and colleagues reports the functional characterization of hevein-like antimicrobial peptides and their derived peptides, with particular regard with their antifungal activity and synergistic activity with other fungicides.

The authors should mention upfront, in the abstract, that these are plant AMPs (not all readers will be familiar with Poaceae).

The introduction is complete and provides a satisfying overview of previous literature on the subject, including specific references to WAMPs. However, I believe a few points should be improved. Namely:

-Lines 58-29: this sentence should be amended, as a large number of known AMPs are not cysteine-rich and do not contain disulfide bonds.

-Very little information if provided about the structure of WAMPs, and this limits the ability of the reader to comprehend the reason why different fragments of these AMPs were tested. Are WAMPs known to undergo proteolytic cleavage in nature? Are they produced as inactive precursors (e.g. as prepropeptides)? Are they typical cationic AMPs? Is their 3D structure known?

All these information can be later extracted from the results section, but it would be wise to anticipate some of the most important data in the introduction. Also, I suggest the addition of a schematic representation of the secondary structure of the peptides below the MSA in Figure 1, along with an indication of disulfide connectivity.

A minor point, which I guess has been already tackled in previous publications. Are the different WAMP variants the product of different paralogous genes or allelic variants of the same locus?

Line 157: “It deserves noting that”. Please use a better wording here.

Lines 166-168: as I mentioned before, adding disulfide connectivity and identifying the gamma-core in figure 1 would be highly beneficial to the readers.

Section 2.3.1: could the authors elaborate a bit the rationale for selecting smaller fragments of a larger AMP for functional characterization? Would not this strategy be expected to disrupt 3D structure due to the loss of disulfide bonds, as discussed in 2.3.3? Hence, what is the rationale for expecting such fragments to retain any antifungal activity whatsoever? Do the authors think that the antifungal activity of WAMPs lies more in their primary sequence than in their 3D structure?

I did not find the description of the methods used for predicting the 3D structure of WAMP-derived peptides satisfying? First and foremost, is a 3D structure determined experimentally through NMR available as a template for modeling? Second, was a homology modeling, fold recognition or ab initio method used? Each predicted 3D structure should be accompanied by structural prediction confidence score.

In light of the aforementioned considerations, it is perhaps not surprising that all the data reported in table 3 and Figure 5 report significant activities only at extremely high (and biologically unrealistic) concentrations. This should be better discussed in this MS.

Although the discussion is comprehensive, I found it to be a little bit too long, and some information concerning the structure and the mode of action of these AMPs could have been easily placed in the introduction section. The authors are advised to revise this section and discuss their results in a more concise way.

Section 4.4: a key point that remains unexplained by the authors is whether a 3D structure of heveins had been previously deposited in PDB or, alternatively, which template has been used for homology modeing. Similarly, no information has been provided about the parameters (and the strategy) used by PEP-FOLD.

Another aspect that would deserve, in my opinion, more space is the discussion about whether the concentrations of these peptides at which a significant biological activity is observed do actually provide any hope for a practical application. I am not a plant expert, but the concentrations reported seem very high to me, and in any case this would raise some concerns about their cytotoxicity, that has not been assessed here (but maybe in some previous works?).

Overall, the English language used is sufficiently clear, but I would suggest the authors to involve a native English speaking colleague in a brief round of revision to polish some sentences that use odd working or contain grammar errors.

Author Response

The manuscript by Odintsova and colleagues reports the functional characterization of hevein-like antimicrobial peptides and their derived peptides, with particular regard with their antifungal activity and synergistic activity with other fungicides.

We are grateful to the Reviewer for thorough analysis of our Manuscript and his valuable comments. Our answers are given below.

 1. The authors should mention upfront, in the abstract, that these are plant AMPs (not all readers will be familiar with Poaceae).

1. We fully agree with the reviewer and added the word “plant” before AMPs in the Abstract on line 6 and emphasized that WAMPs are AMPs found in wheat and other members of Poaceae on lines 9-10.

2. The introduction is complete and provides a satisfying overview of previous literature on the subject, including specific references to WAMPs. However, I believe a few points should be improved. Namely:

-Lines 58-29: this sentence should be amended, as a large number of known AMPs are not cysteine-rich and do not contain disulfide bonds.

2. The Reviewer is absolutely right that among non-plant AMPs, there are numerous families of peptides that do not contain cysteine residues. But it is not the case with plant AMPs. Plant AMPs are cysteine-rich peptides. See, for example the following reviews:

Broekaert, W.F.; Cammue, B.P.A.; De Bolle, M.F.C.; Thevissen, K.; De Samblanx, G.W.; Osborn, R.W. Antimicrobial peptides from plants. Crit. Rev. Plant Sci. 1997, 16, 297–323.  Garcia-Olmedo, F.; Molina, A.; Alamillo, J.M.; Rodriguez-Palenzuela, P. Plant defense peptides. Biopolymers 1998, 47, 479–491.

Manners, J.M. Hidden weapons of microbial destruction in plant genomes. Genome Biol. 2007, 8, 225.

Benko-Iseppon, A.M.; Galdino, S.L.; Calsa, T., Jr.; Kido, E.A.; Tossi, A.; Belarmino, L.C.; Crovella, S. Overview on Plant Antimicrobial Peptides. Curr. Protein Pept. Sci. 2010, 11, 181–188.

Odintsova, T.I.; Egorov, T.A. Plant anti-microbial peptides. In Plant Signaling Peptides; Irving, R., Gehring, C.G., Eds.; Springer: Berlin, Germany, 2012; pp. 107–135. ISBN 978-3-642-27603-3.

 Nawrot, R.; Barylski, J.; Nowicki, G.; Broniarczyk, J.; Buchwald, W.; Go ´zdzicka-Józefiak, A. Plant antimicrobial peptides. Folia Microbiol. 2014, 59, 181–196.

Tam, J.P.; Wang, S.; Wong, K.H.; Tan, W.L. Antimicrobial peptides from plants. Pharmaceuticals 2015, 8, 711–757.

Salas, C.E.; Badillo-Corona, J.A.; Ramírez-Sotelo, G.; Oliver-Salvador, C. Biologically active and antimicrobial peptides from plants. Biomed Res. Int. 2015, 2015, 102129.

Tavormina, P.; De Coninck, B.; Nikonorova, N.; De Smet, I.; Cammue, B.P. The Plant Peptidome: An Expanding Repertoire of Structural Features and Biological Functions. Plant Cell 2015, 27, 2095–2118.

Ageitos, J.M.; Sánchez-Pérez, A.; Calo-Mata, P.; Villa, T.G. Antimicrobial peptides (AMPs): Ancient compounds that represent novel weapons in the fight against bacteria. Biochem. Pharmacol. 2017, 133, 117–138.

Campos, M.L.; De Souza, C.M.; De Oliveira, K.B.S.; Dias, S.C.; Franco, O.L. The Role of Antimicrobial Peptides in Plant Immunity. J. Exp. Bot. 2018, 69, 4997–5011.

Das, K.; Datta, K.; Karmakar, S.; Datta, S.K. Antimicrobial Peptides - Small but Mighty Weapons for Plants to Fight Phytopathogens. Protein Pept. Lett. 2019, 26, 720–742.

All known plant AMP families (defensins, thionins, non-specific lipid-transfer proteins, snakins, hevein-like and kbottin-like peptides, cyclotides) consist of cysteine-rich peptides. Beyond these families, single AMPs isolated from plants do not contain cysteine, such as a glycine- and histidine-rich peptide from Capsella bursa pastoris. However it is not a rule, but an exception.

Park CJ, Park CB, Hong SS, Lee HS, Lee SY, Kim SC.Characterization and cDNA cloning of two glycine- and histidine-rich antimicrobial peptides from the roots of shepherd's purse, Capsella bursa-pastoris. Mol Biol. 2000 Sep;44(2):187-97. doi: 10.1023/a:1006431320677

3. Very little information if provided about the structure of WAMPs, and this limits the ability of the reader to comprehend the reason why different fragments of these AMPs were tested. Are WAMPs known to undergo proteolytic cleavage in nature? Are they produced as inactive precursors (e.g. as prepropeptides)? Are they typical cationic AMPs? Is their 3D structure known? All these information can be later extracted from the results section, but it would be wise to anticipate some of the most important data in the introduction.

 3. (1). Regarding WAMP fragments

One of the objectives of our study was to shed light on the structure-function relationships in the hevein-like peptide WAMPs isolated from wheat seeds and characterized earlier. A common practice in plant AMP research to elucidate the role of different residues/portions of the molecule in the biological activity is to produce ‘mutants” with amino acid substitutions (for example, to replace subsequently each amino acid residue for alanine and test the antimicrobial activity, the so-called alanine scan) or test the activity of short peptides covering the whole amino acid sequence of the native peptide. The latter approach was used in our work and in a number of studies performed by other researchers. See some examples with plant defensins:

Sagehashi, Y.; Takaku, H.; Yatou, O. Partial peptides from rice defensin OsAFP1 exhibited antifungal activity against the rice blast pathogen Pyricularia oryzae. J. Pestic. Sci. 2017, 42, 172–175

Ochiai, A.; Ogawa, K.; Fukuda, M.; Ohori, M.; Kanaoka, T.; Tanaka, T.; Taniguchi, M.; Sagehashi, Y. Rice Defensin OsAFP1 is a New Drug Candidate against Human Pathogenic Fungi. Sci. Rep. 2018, 8, 11434.

Woriedh, M.; Merkl, R.; Dresselhaus, T. Maize EMBRYO SAC family peptides interact differentially with pollen tubes and fungal cells. J. Exp. Bot. 2015, 66, 5205–5216.

Sathoff AE, Velivelli S, Shah DM, Samac DA. Plant Defensin Peptides have Antifungal and Antibacterial Activity Against Human and Plant Pathogens. Phytopathology. 2019 Mar;109(3):402-408. doi: 10.1094/PHYTO-09-18-0331-R.

These studies allowed the authors to identify the regions of the defensin molecule responsible for the antimicrobial activity. To the best of our knowledge, this approach has not been applied to structure-function studies of hevein-like AMPs before our work. We found it interesting and important for the peptide’s mode of action that even short portions of WAMP retained antimicrobial activity which varied depending on the peptide and the fungus.

(2) Regarding the proteolytic cleavage of WAMPs in nature

We showed that WAMPs inhibit fungalysins, secereted metalloproteinases of Fusarium fungi, but are not cleaved by them in vitro:

Slavokhotova, A.A.; Naumann, T.A.; Price, N.P.; Rogozhin, E.A.; Andreev, Y.A.; Vassilevski, A.A.; Odintsova, T.I. Novel mode of action of plant defense peptides - hevein-like antimicrobial peptides from wheat inhibit fungal metalloproteases. FEBS J. 2014, 281, 4754−4764. doi:10.1111/febs.13015.

(3) Regarding biosynthesis

WAMPs are synthesized as precursor proteins containing a signal peptide, a mature peptide and a C-terminal prodomain.

Andreev, Y.A.; Korostyleva, T.V.; Slavokhotova, A.A.; Rogozhin, E.A.; Utkina, L.L.; Vassilevski, A.A.; Grishin, E.V.; Egorov, T.A.; Odintsova, T.I. Genes encoding hevein-like defense peptides in wheat: distribution, evolution, and role in stress response. Biochimie 2012, 94, 1009−1016. doi: 10.1016/j.biochi.2011.12.023.

(4) Regarding 3D structure

The solution structure of WAMP-1a was solved by NMR spectroscopy.

Dubovskii, P.V.; Vassilevski, A.A.; Slavokhotova, A.A.; Odintsova, T.I.;Grishin, E.V.; Egorov, T.A.; Arseniev, A.S. Solution structure of a defense peptide from wheat with a 10-cysteine motif. Biochem. Biophys. Res. Commun. 2011, 411, 14−18. doi: 10.1016/j.bbrc.2011.06.058.

We included the required information concerning WAMPs into the Introduction section on lines 58-66.

4. Also, I suggest the addition of a schematic representation of the secondary structure of the peptides below the MSA in Figure 1, along with an indication of disulfide connectivity.

4. Secondary structure elements are shown in Fig. 3. Disulphide bonds are added to Fig. 1 as recommended and a related mention is added to the Figure caption (line 134)

5. A minor point, which I guess has been already tackled in previous publications. Are the different WAMP variants the product of different paralogous genes or allelic variants of the same locus?

5. WAMPs are products of paralogous genes, not allelic variants of a single gene.

Andreev, Y.A.; Korostyleva, T.V.; Slavokhotova, A.A.; Rogozhin, E.A.; Utkina, L.L.; Vassilevski, A.A.; Grishin, E.V.; Egorov, T.A.; Odintsova, T.I. Genes encoding hevein-like defense peptides in wheat: distribution, evolution, and role in stress response. Biochimie 2012, 94, 1009−1016. doi: 10.1016/j.biochi.2011.12.023.

6. Line 157: “It deserves noting that”. Please use a better wording here.

5. We replaced “It deserves noting that” for “It is noteworthy that” as recommended (line 152 in the revised text).

7. Lines 166-168: as I mentioned before, adding disulfide connectivity and identifying the gamma-core in figure 1 would be highly beneficial to the readers.

7. We designated the disulphide bonds in Fig. 1, gamma-core regions are added in Fig. 3, and related mentions are added to the Figure captions. (Lines 134,185)

8. Section 2.3.1: could the authors elaborate a bit the rationale for selecting smaller fragments of a larger AMP for functional characterization? Would not this strategy be expected to disrupt 3D structure due to the loss of disulfide bonds, as discussed in 2.3.3? Hence, what is the rationale for expecting such fragments to retain any antifungal activity whatsoever? Do the authors think that the antifungal activity of WAMPs lies more in their primary sequence than in their 3D structure?

8. Please see the answer to comment 3.

9. I did not find the description of the methods used for predicting the 3D structure of WAMP-derived peptides satisfying? First and foremost, is a 3D structure determined experimentally through NMR available as a template for modeling? Second, was a homology modeling, fold recognition or ab initio method used? Each predicted 3D structure should be accompanied by structural prediction confidence score.

The solution structure of WAMP-1a was determined by Dubovskii et al. (2011) and loaded into Protein Data Bank (2LB7). We carried out homology modeling WAMPs with substitutions at position 34 using SWISS-MODEL program (Waterhouse et al., 2018), with WAMP-1a as a template.

To quantify modelling errors and give estimates on expected model accuracy, SWISS-MODEL relies on the GMQE and QMEAN scoring function. GMQE (Global Model Quality Estimation) is a quality estimation which combines properties from the target–template alignment and the template structure. The resulting GMQE score is expressed as a number between 0 and 1, reflecting the expected accuracy of a model built with that alignment and template, normalized by the coverage of the target sequence. Higher numbers indicate higher reliability. The QMEAN Z-score provides an estimate of the "degree of nativeness" of the structural features observed in the model on a global scale and is described in Benkert et al. (2011). It indicates whether the QMEAN score of the model is comparable to what one would expect from experimental structures of similar size. QMEAN Z-scores around zero indicate good agreement between the model structure and experimental structures of similar size. Scores of -4.0 or below are an indication of models with low quality.

All predicted 3D structures have GMQE 0.65 and QMEAN: for WAMP-2 -0.97, WAMP-3 -0.73, WAMP-4 -0.90, WAMP-5 -1.00.

The spatial structure of WAMP-derived short peptides was modeled de novo using PEP-FOLD 3 (Lamiable et al., 2016). sOPEP were used to define the best models among several models produced. The best representative models had the lowest sOPEP energy as defined by PEP-FOLD. sOPEP value: for WAMP-N  -10.3467, WAMP-G1  -11.119, WAMP-G2  -14.5086, WAMP-C  -9.28022.

Dubovskii, P.V.; Vassilevski, A.A.; Slavokhotova, A.A.; Odintsova, T.I.;Grishin,E.V.; Egorov, T.A.; Arseniev, A.S. Solution structure of a defense peptide from wheat with a 10-cysteine motif. Biochem. Biophys. Res. Commun. 2011, 411, 14−18. doi:10.1016/j.bbrc.2011.06.058.

Waterhouse, A.; Bertoni, M.; Bienert, S.; Studer, G.; Tauriello, G.; Gumienny, R.; Heer, F.T.; de Beer, T.A.P.; Rempfer, C.; Bordoli, L.; et al. SWISS-MODEL: Homology modelling of protein structures and complexes. Nucleic Acids Res.2018, 46, W296‒W303, doi:10.1093/nar/gky427.

Benkert, P., Biasini, M., Schwede, T. Toward the estimation of the absolute quality of individual protein structure models. Bioinformatics 2011, 27, 343-350.

Lamiable, A.; Thévenet, P.; Rey, J.; Vavrusa, M.; Derreumaux, P.; Tufféry, P. PEP-FOLD3: faster de novo structure prediction for linear peptides in solution and in complex. Nucleic Acids Res. 2016, 44, W449–W454. doi: 10.1093/nar/gkw329.

The necessary changes were made to the section Materials and Methods 4.4 on lines 595-598.

In light of the aforementioned considerations, it is perhaps not surprising that all the data reported in table 3 and Figure 5 report significant activities only at extremely high (and biologically unrealistic) concentrations. This should be better discussed in this MS.

See, please response to comment 12.

10. Although the discussion is comprehensive, I found it to be a little bit too long, and some information concerning the structure and the mode of action of these AMPs could have been easily placed in the introduction section. The authors are advised to revise this section and discuss their results in a more concise way.

10. We included some information on WAMPs in the Introduction section (See, please lines 58-66).

11. Section 4.4: a key point that remains unexplained by the authors is whether a 3D structure of heveins had been previously deposited in PDB or, alternatively, which template has been used for homology modeing. Similarly, no information has been provided about the parameters (and the strategy) used by PEP-FOLD.

11. WAMP-1a (PDB 2LB7) was used as a template for homology modeling of WAMPs with substitutions at position 34.

De novo modeling of WAMP-2-derived peptides using PEP-FOLD 3 was carried out with default parameters. The best structure model for each peptide was chosen based on sOPEP values provided by PEP-FOLD 3 server.

The necessary changes were made to the section Materials and Methods 4.4 on lines 595-598.

12. Another aspect that would deserve, in my opinion, more space is the discussion about whether the concentrations of these peptides at which a significant biological activity is observed do actually provide any hope for a practical application. I am not a plant expert, but the concentrations reported seem very high to me, and in any case this would raise some concerns about their cytotoxicity, that has not been assessed here (but maybe in some previous works?).

12. Regarding practical application of WAMP fragments as fungicides. The reviewer is absolutely right that the concentrations are high, therefore directly as antifungal agents only WAMP-C can be suggested against C. cucumerinum. However (1) The main practical perspective is that these fragments potentiate the effect of a fungicide, and therefore can be used in combination to reduce the negative impact on the environment. Concentrations of WAMP-derived peptides used for sensitization, which seem high, were non-toxic, marginally toxic or low toxic for the pathogens. The low toxicity is one of highly desirable properties of sensitizing agents. A key chemosensitization principle is co-application of such compounds with fungicides, when both fungicide and a sensitizer, applied separately, provide no or insignificant fungitoxic effect (so that not stimulate adaptation mechanisms to a chemical stress in fungi), but effectively suppress a target pathogen at the same concentrations when applied together. This enhances the fungicidal effect and reduces the resistance risk without an increase of fungicide dosages. The search for non-toxic or marginally toxic natural compounds is a general line in our chemosensitization investigations. (2) In principle, the structure of the fragments can be modified to improve antifungal potency. The discovery of the antifungal activity in WAMP-derived fragments mostly contributes to our understanding of the role of different regions of the molecule in the antifungal activity of the hevein-like peptide which is a poorly explored research area.

13. Overall, the English language used is sufficiently clear, but I would suggest the authors to involve a native English speaking colleague in a brief round of revision to polish some sentences that use odd working or contain grammar errors.

13. We showed our manuscript to a native speaker and made the necessary changes and additional editing of fragments of some sentences, (e.g., on lines 38, 39, 41; 47,74, 77, 84-85, 160, 196, 199, 201, 207-209, 212, 231, 232, 242, 243, 272, 302, 310-314, 415, 416, 449, 453-455, 461, 465 ,484, 475, 508). Several grammar errors were found and corrected (e.g., lines 196, 586, 670); the articles, where necessary, were added to nouns throughout the text. All changes (deletions and additions) are highlighted.

Reviewer 2 Report

An interesting study and topic, that demonstrated the practical utility of bioactive compound/drug combination, in which combined subtances potentialize the action of individually low active or inactive compounds.

The study is well conducted, the manuscript is clear and the conclusion is overall adequated.

Comments.

Results section:
How sure are the authors about the correct fold of recombinant WAMPs, as well as the disulfide bond formation in famma-core in WAMP-2-derived peptides? Was it confirmed by MS/MS analysis?

Table 1: Unit is im microMolar, but in the footnote is in microgram mL-1.

Along the text, it is recommended to uniformize the concentration units or keep both together, like microM (microg mL-1)

In figure 5, Table 3 and fig 6-9, it is recommended to keep concentration units in microM (microg mL-1) for clearity and more comprehensive compairison.

Discussion:

In lines 678-679 the authors speculated that binding to nascent chitin is the underlying mechanism of antifungal action of WAMP-derived peptides, particurlay that peptide that has an amino acid extension. How could be this antifungal effect possible?

In lines 681-682, Why did the authors mention that the lipid membrane-disruption dependence for the antifungal effect of wamp-derived peptides is not excludedd? Are there experimental evidences for that (e.g., cytoplasm extravasion, cell-imperment dye uptake, hydrolytic enzyme release, etc)?

It would be recommended to authors disccus about the mechanism of action of tubuconazole, for the readers get insights on the synergistic effect of tubuconazole and WAMP-peptide derivates

The last sentence, line 694, should be connected to the hypothesis of chitin-binding-dependent action and membrane disruptive action, since these mechanisms were speculated in the discussion.

Author Response

An interesting study and topic, that demonstrated the practical utility of bioactive compound/drug combination, in which combined subtances potentialize the action of individually low active or inactive compounds.

The study is well conducted, the manuscript is clear and the conclusion is overall adequated.

We are grateful to the Reviewer for thorough analysis of our Manuscript and his valuable comments and recommendations. Our answers are given below.

Comments.
Results section:
1. How sure are the authors about the correct fold of recombinant WAMPs, as well as the disulfide bond formation in famma-core in WAMP-2-derived peptides? Was it confirmed by MS/MS analysis?

1. Recombinant production of WAMPs was described in detail in our papers:

Odintsova TI, Vassilevski AA, Slavokhotova AA, Musolyamov AK, Finkina EI, Khadeeva NV, Rogozhin EA, Korostyleva TV, Pukhalsky VA, Grishin EV, Egorov TA. A novel antifungal hevein-type peptide from Triticum kiharae seeds with a unique 10-cysteine motif. FEBS J. 2009 Aug;276(15):4266-75. doi: 10.1111/j.1742-4658.2009.07135.x.

Slavokhotova AA, Naumann TA, Price NP, Rogozhin EA, Andreev YA, Vassilevski AA, Odintsova TI. Novel mode of action of plant defense peptides - hevein-like antimicrobial peptides from wheat inhibit fungal metalloproteases. FEBS J. 2014 Oct;281(20):4754-64. doi: 10.1111/febs.13015.

The peptides were produced as fusion proteins with thioredoxin, providing proper folding and correct disulfide bond formation (LaVallie ER, DiBlasio EA, Kovacic S, Grant KL, Schendel PF & McCoy JM (1993) A thioredoxin gene fusion expression system that circumvents inclusion body formation in the E. coli cytoplasm. Biotechnology (NY) 11, 187–193). The molecular masses of the recombinant peptides determined by MALDI-TOF MS were equal to the expected masses of the peptides with all 10 cysteine residues being engaged in disulphide bonds. Furthermore, the recombinant peptide WAMP-1b co-eluted with the native WAMP-1b isolated from seeds and had the same retention time when analyzed by analytical RP-HPLC, it also had the expected N-terminal amino acid sequence as determined by direct Edman sequencing.

WAMP-2-derived peptides were synthesized in a non-reduced form as shown by MALDI-TOF MS as it was mentioned on lines 589-590 of the revised text (The identity of the synthesized peptides was verified by mass spectrometry. The purity of peptides was ˃95%.)

2. Table 1: Unit is im microMolar, but in the footnote is in microgram mL-1.
Along the text, it is recommended to uniformize the concentration units or keep both together, like microM (microg mL-1). In figure 5, Table 3 and fig 6-9, it is recommended to keep concentration units in microM (microg mL-1) for clearity and more comprehensive compairison.

We converted all IC50 units given in µM into µg/mL, as recommended.

Discussion:
3. In lines 678-679 the authors speculated that binding to nascent chitin is the underlying mechanism of antifungal action of WAMP-derived peptides, particurlay that peptide that has an amino acid extension. How could be this antifungal effect possible?

3. The hypothesis that hevein-like AMPs inhibit fungal growth via binding to nascent chitin chains is widely accepted by the researchers involved in studies of plant hevein-like AMPs. See for example:

Jan Van Parijs WFB, Irwin J. Goldstein , and Willy J. Peumans Hevein an Antifungal Protein from Rubber-Tree. Planta 1990, 258-264; Van den Bergh KP, Rouge P, Proost P, Coosemans J, Krouglova T,

Engelborghs Y, Peumans WJ, Van Damme EJ: Synergistic Antifungal Activity of Two Chitin-Binding Proteins from Spindle Tree (Euonymus Europaeus L.). Planta 2004, 219(2):221-232).

Binding to chitin is mediated by three aromatic residues of the chitin-binding site present in WAMP-G2 and partially in WAMP-G1, so in lines 678-679, binding to chitin as an antifungal mechanism was speculated for WAMP-G2.

4. In lines 681-682, Why did the authors mention that the lipid membrane-disruption dependence for the antifungal effect of wamp-derived peptides is not excludedd? Are there experimental evidences for that (e.g., cytoplasm extravasion, cell-imperment dye uptake, hydrolytic enzyme release, etc)?

The elucidation of the mode of action of WAMP-2-derived peptides was beyond the scope of the present study. We have just hypothesized that membrane disruption could be a possibility. This follows from our experimental evidence that they potentiate the effect of the fungicide. However, the exact molecular mechanisms should be further explored and we are going to do it in our future studies.

5. It would be recommended to authors disccus about the mechanism of action of tubuconazole, for the readers get insights on the synergistic effect of tubuconazole and WAMP-peptide derivates

5. We added more detailed information on triazole mode of action and edited the fragment to emphasize the different targets of the fungicide and WAMPs on lines 102-105.

6. The last sentence, line 694, should be connected to the hypothesis of chitin-binding-dependent action and membrane disruptive action, since these mechanisms were speculated in the discussion.

6. The last sentence of the Conclusion Section line 694 is as follows: “The mechanisms underlying the sensitizing activity of WAMP-2 fragments remain so far unknown; they will be the subject of further investigations”. We believe that since these mechanisms have not been studied in this paper it seems unreasonable to mention our considerations not substantiated by experimental evidence in conclusions.

Reviewer 3 Report

In this study, Odintsova et al. examine the structure-activity relationships of the hevein-like antimicrobial peptides WAMPS from plants belonging to the family Poaceae.  WAMPs are small cysteine-rich peptides containing 10 cysteines that are presumably involved in the formation of pentadisulfide array. WAMPs are specific inhibitors of fungalysin, a secreted metalloproteinase of Fusarium fungi that specifically cleaves plant chitinases between the chitin-binding and catalytic domains, thus enabling plants to degrade fungal chitin and protect themselves. The authors have determined the antifungal activity of five WAMP homologs that differ in the identity of the amino acid residue at 34.  They show that these homologs inhibit spore germination in several fungal pathogens, but with significantly differing potency. This result is not surprising sine these homologs differ in net charge and hydrophobicity.  The authors further analyze the antifungal activity of the four fragments of WAMP-2 peptide and document that the N-terminal fragment and the C-terminal fragment possess significantly more antifungal activity than the fragments WAMP-G1 and WAMPG2 fragments.  The novel finding of the study is that the fragments of this peptide synergize the antifungal activity of the fungicide tebuconazole which inhibits ergosterol biosynthesis in fungal pathogens.

The authors need to address the following issues:

  1. It is not stated anywhere in the manuscript what WAMP stands for. 
  2. Authors have not provided any evidence in the manuscript that WAMP peptides expressed in E. coli are correctly folded with the expected disulfide bond arrangement. 
  3. The gamma-core motifs in WAMP-G1 (CGKYGFC) and in WAMP-G2  (GFCGSGDAYC) are really not bona fide gamma core motifs. The typical gamma-core motif of an antimicrobial peptide has the consensus sequence GXCX3-9C and contains at least one positively charged amino acid and one or more hydrophobic residues. The authors should remove these gamma-core motifs in the manuscript. 
  4. The C-terminal fragment of WAMP-2 peptide contains the gamma-core motif CQSQCRG, but has not been mentioned in the manuscript. 
  5. It is not clear why the authors did not test the ability of the whole WAMP-2 peptide to sensitize the antifungal activity of tebuconazole.
  6. WAMP-G2 peptide contains three aromatic amino acid residues involved in chitin binding.  Have the authors determined whether or not WAMP-G2 peptide binds to chitin?
  7. The authors conclude that some WAMP peptides inhibit fungal spore germination through mechanisms other than chitin binding.  Have they determined if one or more of these peptides permeabilize the plasma membrane of a fungal pathogen used in the study?      

Author Response

In this study, Odintsova et al. examine the structure-activity relationships of the hevein-like antimicrobial peptides WAMPS from plants belonging to the family Poaceae.  WAMPs are small cysteine-rich peptides containing 10 cysteines that are presumably involved in the formation of pentadisulfide array. WAMPs are specific inhibitors of fungalysin, a secreted metalloproteinase of Fusarium fungi that specifically cleaves plant chitinases between the chitin-binding and catalytic domains, thus enabling plants to degrade fungal chitin and protect themselves. The authors have determined the antifungal activity of five WAMP homologs that differ in the identity of the amino acid residue at 34.  They show that these homologs inhibit spore germination in several fungal pathogens, but with significantly differing potency. This result is not surprising sine these homologs differ in net charge and hydrophobicity.  The authors further analyze the antifungal activity of the four fragments of WAMP-2 peptide and document that the N-terminal fragment and the C-terminal fragment possess significantly more antifungal activity than the fragments WAMP-G1 and WAMPG2 fragments.  The novel finding of the study is that the fragments of this peptide synergize the antifungal activity of the fungicide tebuconazole which inhibits ergosterol biosynthesis in fungal pathogens.

We are grateful to the Reviewer for thorough analysis of our Manuscript and his valuable comments. Our answers are given below.

The authors need to address the following issues:

 1. It is not stated anywhere in the manuscript what WAMP stands for.

1. WAMP stands for Wheat Antimicrobial Peptide. The explanation was given in our manuscript (Odintsova et al., 2009), which describes primary structure determination and antimicrobial activity of the peptide.

Odintsova TI, Vassilevski AA, Slavokhotova AA, Musolyamov AK, Finkina EI, Khadeeva NV, Rogozhin EA, Korostyleva TV, Pukhalsky VA, Grishin EV, Egorov TA. A novel antifungal hevein-type peptide from Triticum kiharae seeds with a unique 10-cysteine motif. FEBS J. 2009 Aug; 276(15):4266-75. doi: 10.1111/j.1742-4658.2009.07135.x.

We added the designation of the abbreviation WAMP to the Abstract. (lines 9-10).

2. Authors have not provided any evidence in the manuscript that WAMP peptides expressed in  coliare correctly folded with the expected disulfide bond arrangement. 

2. Recombinant production of WAMPs was described in detail in our papers:

Odintsova TI, Vassilevski AA, Slavokhotova AA, Musolyamov AK, Finkina EI, Khadeeva NV, Rogozhin EA, Korostyleva TV, Pukhalsky VA, Grishin EV, Egorov TA. A novel antifungal hevein-type peptide from Triticum kiharae seeds with a unique 10-cysteine motif. FEBS J. 2009 Aug;276(15):4266-75. doi: 10.1111/j.1742-4658.2009.07135.x.

Slavokhotova AA, Naumann TA, Price NP, Rogozhin EA, Andreev YA, Vassilevski AA, Odintsova TI. Novel mode of action of plant defense peptides - hevein-like antimicrobial peptides from wheat inhibit fungal metalloproteases. FEBS J. 2014 Oct;281(20):4754-64. doi: 10.1111/febs.13015.

The peptides were produced as fusion proteins with thioredoxin, providing proper folding and correct disulfide bond formation (LaVallie ER, DiBlasio EA, Kovacic S, Grant KL, Schendel PF & McCoy JM (1993) A thioredoxin gene fusion expression system that circumvents inclusion body formation in the E. coli cytoplasm. Biotechnology (NY) 11, 187–193).

Furthermore, the recombinant peptide WAMP-1b co‐eluted with the native WAMP-1b and had the same retention time when analyzed by analytical RP‐HPLC, it also had the expected N‐terminal amino acid sequence as determined by direct Edman sequencing. The molecular mass of the recombinant product obtained by MALDI MS was equal to the mass measured for the native WAMPs.

3. The gamma-core motifs in WAMP-G1 (CGKYGFC) and in WAMP-G2  (GFCGSGDAYC) are really not bona fide gamma core motifs. The typical gamma-core motif of an antimicrobial peptide has the consensus sequence GXCX3-9C and contains at least one positively charged amino acid and one or more hydrophobic residues. The authors should remove these gamma-core motifs in the manuscript. 

3. Gamma-core motifs were identified in WAMP structure according to the definition of Yount and Yeaman (2004) (Yount, N.Y.; Yeaman, M.R. Multidimensional signatures in antimicrobial peptides. Proc. Natl Acad. Sci. USA 2004, 101, 7363–7368). In addition to the amino acid sequence motif, it implies the presence of two beta-strands connected by a short loop in the three-dimensional structure. WAMP-G1 (CGKYGFC) and in WAMP-G2 (GFCGSGDAYC) fulfill this requirement, while WAMP-C does not. The selection of WAMP regions corresponding to the gamma-core is described in the Results Section 2.3.1.

4. The C-terminal fragment of WAMP-2 peptide contains the gamma-core motif CQSQCRG, but has not been mentioned in the manuscript. 

4. In the WAMP-2 structure, CQSQCRG does not adopt a conformation with two beta-strands and a connecting loop (see above).

5. It is not clear why the authors did not test the ability of the whole WAMP-2 peptide to sensitize the antifungal activity of tebuconazole.

6. Thank you for calling our attention to this issue. Although we found previously a very insignificant sensitizing activity of native WAMP-2 for some microorganisms, this peptide did not show it for the tested wheat pathogens. According to the recommendation, the lack of activity is mentioned now on lines 622-623. 

The findings on a higher biological activity of peptide fragments compared to a native peptide are not so surprising. There are also reports that the protective activity of a short peptide fragment tested against plant pathogens can be the same or even higher than that of a source peptide, e.g., peptides derived from bacterial flagellin, or see a reference below.

We did not aim to study WAMP-2 activity also because the chemical synthesis of the complete peptide or the production of recombinant WAMP-2 by heterologous expression in quantities sufficient for plant treatments are too complex and expensive for production of compounds for agricultural usage. At the same time, short peptides can be produced by almost quantitatively unlimited and rather low-cost chemical synthesis.

We specified the goals of production and the test scope of WAMP-2 and other homologues on lines 583-584.

Dzhavakhiya, V.G. et al., Search for the active center of peptidyl-prolyl cys/trans isomerase from Pseudomonas fluorescens responsible for the induction of tobacco ( Nicotiana tabacum L.) plant resistance to tobacco mosaic virus. Agricultural Biology (Sel’skokhozyaistvennaya biologiya), 2016, 51 (3), 392-400. doi: 10.15389/agrobiology.2016.3.392rus.

6. WAMP-G2 peptide contains three aromatic amino acid residues involved in chitin binding.  Have the authors determined whether or not WAMP-G2 peptide binds to chitin?

6. No, we have not. We are going to do it in our future studies.

7. The authors conclude that some WAMP peptides inhibit fungal spore germination through mechanisms other than chitin binding.  Have they determined if one or more of these peptides permeabilize the plasma membrane of a fungal pathogen used in the study?

7. No, we have not. We are going to do it in the following experiments.

Reviewer 4 Report

The paper describes Hevein-like peptides antifungal activity and their  sensitization of plant pathogenic fungi to tebuconazole. The manuscript is well written and contains new and interesting information about the effects of these peptides on certain phytopathogenic fungi.

Major comments:

In my opinion the manuscript is well designed but too long and I recommend its shortening.

In section 2. Results - I suggest to remove the fragments (e.g line 112-116) or transfer these fragments to Discussion section. In this section only Results should be described.

In section 3. Discussion - I also suggest to shorten this part. In its current form, it is a mini-review, not a discussion of the obtained results.

Specific comments:

  1. Table 1 – please provide Statistical Analysis for this results
  2. Line 315 – delete extra space
  3. Figure 7 – 9 – Please provide full names of the fungal species on graphs. Introducing shortcuts of fungal names is confusing.
  4. Figure 6 – please change colors of the bars. The colors are too similar.
  5. Table 4 – FFCIs – introduce to the table full  name "fractional fungicidal concentration indices" not only an abbreviation expansion.
  6. Line 648, 656 – in the text are some parts highlighted in yellow. Why?
  7. Line 690 – tubuconazole change to tebuconazole
  8. Please check carefully the References section.

Author Response

The paper describes Hevein-like peptides antifungal activity and their  sensitization of plant pathogenic fungi to tebuconazole. The manuscript is well written and contains new and interesting information about the effects of these peptides on certain phytopathogenic fungi.

We are grateful to the Reviewer for his valuable comments and recommendations. Our answers are given below.

Major comments:

In my opinion the manuscript is well designed but too long and I recommend its shortening.

In section 2. Results - I suggest to remove the fragments (e.g line 112-116) or transfer these fragments to Discussion section. In this section only Results should be described.

We transfered these lines to the Introduction (Lines 61-65).

In section 3. Discussion - I also suggest to shorten this part. In its current form, it is a mini-review, not a discussion of the obtained results.

We think that information concerning plant hevein-like peptides is of interest to those who are involved in AMPs research. Taking into account that only a handful of hevein-like peptides were studied, we decide to leave the Discussion as it is.

Specific comments:

1. Table 1 – please provide Statistical Analysis for this results

1. In Table 1, standard deviations are shown. We added the necessary information in line 151.

2. Line 315 – delete extra space

2. The extra space under Fig. 6 is deleted.

3. Figure 7 – 9 – Please provide full names of the fungal species on graphs. Introducing shortcuts of fungal names is confusing.

3. Full pathogen names are added to the captions under figures 7-9.

4. Figure 6 – please change colors of the bars. The colors are too similar.

4. We altered one of colors to make the bars more contrast.

5. Table 4 – FFCIs – introduce to the table full  name "fractional fungicidal concentration indices" not only an abbreviation expansion.

We would prefer to leave the abbreviation in the table. The full name and corresponding abbreviation are given in the table name. We have now written the first letters in the full title in capital letters. The introduction of the full term instead of FFCIs increases the table size and makes it very difficult to format subsequent sections: captions and explanations to figures cannot be placed on the same page as the figure; this is a violation of MFPI's manuscript design style.

6. Line 648, 656 – in the text are some parts highlighted in yellow. Why?

6. We don’t know. For verification, we uploaded the submitted article. There are no highlighting in yellow on lines 648, 656 and throughout the text.

7. Line 690 – tubuconazole change to tebuconazole (now line 670)

7. Thank you. This is corrected.

8. Please check carefully the References section.

8. This section is checked and some mistakes are corrected according to MDPI’s style for references lists and the ACS Style Guide.

Round 2

Reviewer 1 Report

Thank you for providing a revised version of your manuscript. I feel like most of my concerns have been adequately solved.

Reviewer 4 Report

The paper was corrected according to my sugestions. My recommendation is to accept it in the present form.